# Personalization of Large Language Models: A Survey

**Zhehao Zhang**[1], **Ryan A. Rossi**[2], **Branislav Kveton**[2], **Yijia Shao**[3], **Diyi Yang**[3],
**Hamed Zamani**[4], **Franck Dernoncourt**[2], **Joe Barrow**[5], **Tong Yu**[2], **Sungchul Kim**[2],
**Ruiyi Zhang**[2], **Jiuxiang Gu**[2], **Tyler Derr**[6], **Hongjie Chen**[7], **Junda Wu**[8], **Xiang Chen**[2],
**Zichao Wang**[2], **Subrata Mitra**[2], **Nedim Lipka**[2], **Nesreen Ahmed**[9], **Yu Wang**[10]

[1]*Dartmouth College*
[2]*Adobe Research*
[3]*Stanford University*
[4]*University of Massachusetts Amherst*
[5]*Pattern Data*
[6]*Vanderbilt University*
[7]*Dolby Research*
[8]*University of California San Diego*
[9]*Cisco Research*
[10]*University of Oregon*

**Reviewed on OpenReview:** *https://openreview.net/forum?id=tf6A9EYMo6*

## Abstract

Personalization of Large Language Models (LLMs) has recently become increasingly important with a wide range of applications. Despite the importance and recent progress, most existing works on personalized LLMs have focused either entirely on (a) personalized text generation or (b) leveraging LLMs for personalization-related downstream applications, such as recommendation systems. In this work, we bridge the gap between these two separate main directions for the first time by introducing a taxonomy for personalized LLM usage and summarizing the key differences and challenges. We provide a formalization of the foundations of personalized LLMs that consolidates and expands notions of personalization of LLMs, defining and discussing novel facets of personalization, usage, and desiderata of personalized LLMs. We then unify the literature across these diverse fields and usage scenarios by proposing systematic taxonomies for the granularity of personalization, personalization techniques, datasets, evaluation methods, and applications of personalized LLMs. Finally, we highlight challenges and important open problems that remain to be addressed. By unifying and surveying recent research using the proposed taxonomies, we aim to provide a clear guide to the existing literature and different facets of personalization in LLMs, empowering both researchers and practitioners.

## 1 Introduction

Large language models (LLMs) have emerged as powerful tools capable of performing a wide range of natural language processing (NLP) tasks with remarkable proficiency (*e.g.*, Radford et al., 2018; Devlin et al., 2019; Lewis et al., 2019; Radford et al., 2019; Brown et al., 2020; Raffel et al., 2020; Achiam et al., 2023; Touvron et al., 2023; Groeneveld et al., 2024). Empirically, these models have demonstrated their capability as generalist models, allowing them to perform numerous tasks such as text generation, translation, summarization, and question-answering with decent accuracy. Notably, LLMs can perform effectively in zero-shot or few-shot settings, meaning they can follow human instructions and perform complex tasks with little to no task-specific training data (Bommasani et al., 2021; Liu et al., 2023c). This capability eliminates the need for extensive fine-tuning of their parameters, thereby significantly simplifying human interaction with machines through straightforward input prompts. For instance, users can engage with LLMs in a

conversational format, making interactions more intuitive and accessible. Such robust and versatile abilities of LLMs have led to the creation of numerous applications, including general AI assistants (AutoGPT, 2024), copilots (Microsoft, 2024), and personal LLM-based agents (Li et al., 2024h). These applications assist users in a wide range of activities such as writing emails, generating code, drafting reports, and more.

Recently, there has been growing interest in adapting LLMs to user-specific contexts, beyond their natural use as NLP task solvers or general-purpose chatbots (Tseng et al., 2024). To this end, personalization of LLMs addresses this by adapting the models to generate responses that cater to the unique needs and preferences of each user or user group (Salemi et al., 2023). Such personalization is crucial for human-AI interaction and user-focused applications. It is expected to enhance user satisfaction by providing more relevant and meaningful interactions, ensuring users receive responses that are more aligned with their needs and expectations. This enables LLMs to offer more effective assistance across a diverse range of applications such as customer support (Amazon, 2024), where personalized responses can significantly improve user experience; education (Wang et al., 2022; 2024b), where tailored content can better meet individual learning needs (Woźniak et al., 2024); and healthcare, where personalized advice can enhance patient care (Tang et al., 2023; Yuan et al., 2023).

Personalization of LLMs has recently attracted significant attention (Salemi et al., 2023; Tseng et al., 2024), with research primarily focusing on two directions: (a) personalized text generation, which tailors generated text to user-specific contexts, and (b) downstream task personalization, which leverages LLM capabilities to improve performance on targeted applications such as recommendation systems. Despite the extensive research efforts, these two areas have historically developed independently due to technical limitations and methodological differences, often resulting in existing surveys (Chen, 2023; Chen et al., 2024b;c) examining each aspect in isolation. However, these two domains are not fundamentally distinct, as LLMs possess the flexibility to adapt to a wide range of tasks (Qin et al., 2023). Rather, as LLM capabilities continue to evolve, we envision that those two directions will increasingly converge, enabling unified systems in which a single intelligent agent seamlessly transitions from engaging in personalized conversations to reasoning over structured knowledge like product catalogs for task-oriented recommendations. Bridging this current conceptual gap by synthesizing insights across both dimensions thus constitutes an essential step toward creating fully integrated, adaptable, and generalizable user experiences.

To fully understand LLM personalization, it is important to examine these research directions within a unified framework that captures the broader landscape of personalization. Beyond integrating different approaches, a more in-depth discussion is needed on the foundational concepts, techniques, datasets, and evaluation methods that support LLM personalization. Additionally, real-world challenges such as balancing personalization with privacy concerns, mitigating biases, and handling data limitations must be addressed to ensure practical and ethical deployment. In this survey, we provide a comprehensive perspective by introducing systematic taxonomies that categorize personalization based on granularity, methodology, and evaluation strategies. We also explore open problems and potential research opportunities. Through this framework, we connect the various aspects of personalization and offer a structured reference that outlines the essential components needed to develop and evaluate personalized LLMs.

The key contributions of this work are as follows:

1. **A unifying view and taxonomy for the usage of personalized LLMs (Section 2).** We provide a unifying view and taxonomy of the usage of personalized LLMs based on whether they focus on evaluating the generated text directly, or whether the text is used indirectly for another downstream application. This serves as a fundamental basis for understanding and unifying the two separate areas focused on the personalization of LLMs. Further, we analyze the limitations of each, including the features, evaluation, and datasets, among other factors.
2. **A formalization of personalized LLMs (Section 3).** We provide a formalization of personalized LLMs by establishing foundational concepts that consolidate existing notions of personalization, defining and discussing novel facets of personalization, and outlining desiderata for their application across diverse usage scenarios.
3. **An analysis and taxonomy of the personalization granularity of LLMs (Section 4).** We propose three different levels of personalization granularity for LLMs, including (i) user-level person-

alization, (ii) persona-level personalization, and (iii) global preference personalization. We formalize these levels, and then discuss and characterize the trade-offs between the different granularities of LLM personalization. Notably, user-level personalization is the finest granularity; however, it requires a sufficient amount of user-level data. In contrast, persona-level personalization groups users into personas and tailors the experience based on persona assignments. While it doesn't provide the same granularity as user-level personalization, it is effective for personalizing experiences for users with limited data. Finally, global personalization caters to the overall preferences of the general public and does not offer user-specific personalization.[1]

4. **A survey and taxonomy of techniques for LLM personalization (Section 5).** We categorize and provide a comprehensive overview of the current techniques for personalizing LLMs based on how user information is utilized. Our taxonomy covers various categories of methods such as retrieval-augmented generation (RAG), prompt engineering, supervised fine-tuning, embedding learning, and reinforcement learning from human feedback (RLHF). For each category of methods, we discuss their unique characteristics, applications, and the trade-offs involved. Our detailed analysis helps in understanding the strengths and limitations of different personalization techniques and their suitability for various tasks.

5. **A survey and taxonomy of metrics and evaluation of personalized LLMs (Section 6).** We categorize and analyze the existing metrics used for evaluating personalized LLMs, proposing a novel taxonomy that distinguishes between direct and indirect evaluation methods. We highlight the importance of both qualitative and quantitative metrics, addressing various facets such as user satisfaction, relevance, and coherence of the generated text. Additionally, we discuss the challenges in evaluating personalized LLMs and suggest potential solutions to improve the robustness and reliability of the evaluation process.

6. **A survey and taxonomy of datasets for personalized LLMs (Section 7).** We provide a comprehensive taxonomy of datasets used for training and evaluating personalized LLMs, categorizing them based on their usage in direct or indirect evaluation of personalized text generation. Our survey covers a wide range of datasets, including those specifically designed for short- and long-text generation, recommendation systems, classification tasks, and dialogue generation. We discuss the strengths and limitations of each dataset, their relevance to different personalization techniques, and the need for more diverse and representative datasets to advance the field.

7. **A survey of applications for personalized LLMs (Section 8).** We survey key domains where personalized LLMs are applied, including AI assistants in education and healthcare, finance, legal, and coding environments. We also explore their use in recommendation systems and search engines, highlighting the ability of personalized LLMs to deliver customized user experiences, enhance engagement, and improve task-specific outcomes across diverse fields.

8. **An overview of important open problems and challenges for future work to address (Section 9).** We outline critical challenges and open research questions in personalized LLMs that need to be addressed for advancing the field. Key issues include the need for improved benchmarks and metrics to evaluate personalization effectively, tackling the cold-start problem in adapting models to sparse user data and addressing stereotypes and biases that may arise in personalized outputs. Privacy concerns surrounding user-specific data are also explored, particularly in balancing personalization with privacy protection. Additionally, we discuss the unique complexities of expanding personalization to multi-modal systems, where integrating user preferences across diverse input types remains an open challenge.

In the remainder of the article, we first present a unifying view and taxonomy for the usage of personalized LLMs (Section 2), and then delve into the theoretical foundations of personalized LLMs (Section 3). Next, we explore the granularity of personalization in LLMs (Section 4), and provide a comprehensive survey and taxonomy of techniques for personalized LLMs (Section 5). We then categorize metrics and methods for the evaluation of personalized LLMs (Section 6), and offer a detailed taxonomy of datasets used for personalized LLMs (Section 7). We discuss the various applications of personalized LLMs (Section 8), and finally, identify key challenges and propose future research directions (Section 9).

---

[1]We include it here for completeness, though it is not the focus of this work.

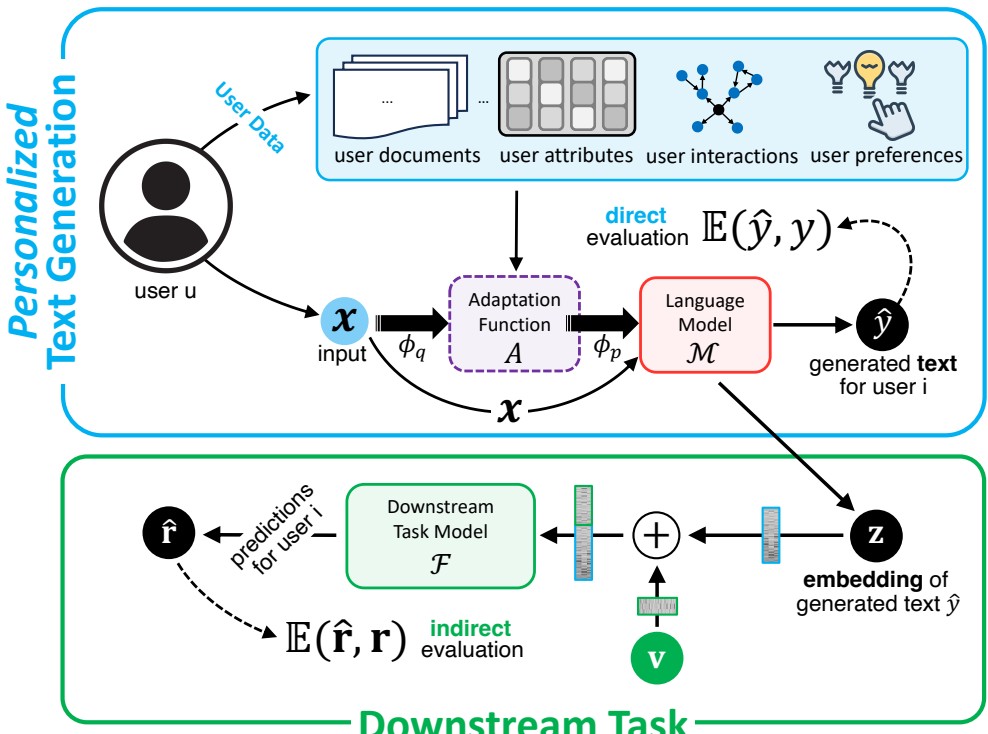

Figure 1: **Taxonomy for Personalized LLM Usage.** To bridge the gap in the existing literature on personalized LLMs, we propose the intuitive taxonomy outlined above, which categorizes work into two main areas. The first focuses on studying the ❶ **personalized text generated** directly, while the second emphasizes using personalized information as intermediate steps or implicitly as embeddings to improve the quality of a ❷ **downstream task** such as recommendation systems. See Section 2 for a detailed discussion. An example of an adaptation function $A$ is a retrieval module. Please note that $y$ here can represent user-written text if available, or alternatively, user preferences and separate reward models that reflect user judgments.

## 2 Unifying Personalized LLMs

To bridge the gap between the two distinct lines of work in the literature, we propose an intuitive taxonomy (see Figure 1) that categorizes personalized LLM efforts into two main categories: ❶ *personalized text generation*, and ❷ *downstream task personalization*. In the first category, *personalized text generation*, the goal is to generate text that directly aligns with individual or group preferences (Salemi et al., 2023; Kumar et al., 2024). For example, a personalized mental health chatbot should generate empathetic responses based on a user's previous conversations, adapting the tone and language to reflect their emotional state. The focus is on producing personalized content, which is evaluated by assessing the quality of the generated text itself using user-written text if available, or alternatively, user preferences and separate reward models that reflect user judgments, as the generated text should match or approximate the style or content the user would produce. In the second category, *downstream task personalization*, personalized LLMs are used to enhance the performance of a specific task, such as recommendation (Lyu et al., 2023; Bao et al., 2023). For instance, an LLM-enhanced movie recommendation system might suggest new films by analyzing a user's viewing history, preferences, and interactions with previous recommendations. In this scenario, the LLM may generate intermediate tokens or embeddings that enhance the system's performance on specific downstream tasks. While these intermediate tokens are not evaluated directly, they serve as crucial steps in improving the overall effectiveness of the task-specific system. The performance is assessed through task-specific metrics like recommendation accuracy or task success rates. Unlike the first category, this line of work focuses on improving task outcomes rather than the text generation process.

DIRECT PERSONALIZED TEXT GENERATION: While there are various techniques for generating personalized text from LLMs, We introduce a general adaptation function $A$ that integrates user-specific information into personalized text generation. An example of $A$ is a retrieval module, with additional concrete examples provided in Figure 7. For a given user $u$, this information may include user-written documents, static attributes, interaction histories, or preferences, as illustrated in Figure 1. Given a user's textual input $x$, a query generation function $\phi_q$ transforms this input to effectively integrate relevant user-specific data via the adaptation function $A$. The adaptation function $A$ leverages the transformed query $\phi_q(x)$, user data $\mathcal{D}_u$, and optionally a parameter $k$ for flexibility. Subsequently, another transformation function, the personalized prompt generation function $\phi_p$, combines the original input $x$ and the output of the adaptation function to form a personalized input $\bar{x}$. Ultimately, the personalized text $\hat{y}$ generated by an LLM $\mathcal{M}$ is:

$$\hat{y} = \mathcal{M}(\phi_p(x, \mathcal{A}(\phi_q(x), \mathcal{D}_u, k))) = \mathcal{M}(\bar{x}) \tag{1}$$

Some existing approaches focus exclusively on generating this personalized text $\hat{y}$ and then directly evaluating its quality. Evaluation typically involves comparing $\hat{y}$ to some form of ground truth or user-specific reference, denoted as $y$. The reference $y$ may represent actual user-written text when available, or more generally, it may encompass user preferences or evaluations obtained from dedicated reward models or user judgments. The evaluation is performed using a metric $\mathbb{E}(\hat{y}, y)$, such as ROUGE-1, ROUGE-L, METEOR, or other specialized metrics designed for personalized text generation tasks.

While evaluating how well the personalized text generated, $\hat{y}$, matches the known user preferences is crucial, it remains particularly challenging due to the scarcity of datasets with high-quality, user-written labels. This likely contributes to the limited focus on the foundational task of personalized text generation in the literature. Instead, there have been many more works that focus on utilizing the personalized text generated $\hat{y}$ *indirectly* to improve a downstream task such as recommendation or prediction in general. It is important to note that in these indirect approaches, where the focus is on improving a downstream task, the personalized text output $\hat{y}$ is typically not evaluated and is considered less critical. The key consideration is that the generated intermediate text or its embedding when applied to a downstream task, can enhance the overall system's performance. While this line of work lacks interpretability regarding the intermediate information generated by LLMs, it has been demonstrated that augmenting systems with this information typically improves performance in downstream personalization-related applications. For a detailed discussion of the evaluation specific to personalized LLMs, see Sec. 6.

INDIRECT DOWNSTREAM TASK: Instead of studying how to directly generate text $\hat{y}$ generated for user $i$, many works focus on leveraging $\hat{y}$ or its personalized embedding $\mathbf{z}$ to improve downstream tasks, such as recommendation. Figure 1 provides an intuitive overview of the fundamental steps used in these approaches. Typically, these methods utilize the embedding $\mathbf{z}$ or $\hat{y}$ as additional information and augment it with other information relevant to the downstream task. In Figure 1, the user-specific embedding $\mathbf{z}$ or intermediate text $\hat{y}$ is augmented with another embedding or task-specific text $\mathbf{v}$ (e.g., concatenated or combined using a function) to form a unified representation that is then passed into the downstream task model $\mathcal{F}$, which can represent any model for a specific application, such as a recommendation system. Although Figure 1 shows a single embedding or text $\mathbf{v}$ combined with $\mathbf{z}$ or $\hat{y}$, in practice, multiple ones or hierarchical combinations can be applied. The downstream model $\mathcal{F}$ then produces predictions $\hat{\mathbf{r}}$, which could include inferred ratings or scores, among other outputs.

While direct ❶ *personalized text generation* and ❷ *downstream task personalization* might appear distinct, they share many underlying components and mechanisms. Both settings often involve retrieving and utilizing user-specific data, constructing personalized prompts or embeddings, and leveraging these to enhance model outputs. The key distinction lies in the dataset they use and the evaluation methods: direct text generation focuses on aligning the generated text with user-written ground-truth, while downstream task personalization evaluates the improvement in specific tasks. Despite these differences, the two approaches can complement each other. For instance, advancements in direct personalized text generation can provide richer, more nuanced intermediate text or embeddings that may enhance downstream tasks. Conversely, improvements in downstream task personalization models can inform better methods for retrieving and leveraging user-specific data in direct generation tasks. By viewing both approaches as two sides of the same coin, researchers from

these communities can benefit from cross-pollination. This unification offers an opportunity to share best practices, datasets, and techniques across the two lines of work, driving progress in both areas. In the next section, we delve into these shared foundations, laying out the core principles and formal definitions that unify both lines of work. By framing personalization in a comprehensive theoretical context, we aim to establish a shared vocabulary and methodology that can facilitate cross-disciplinary collaboration between these communities, fostering new insights and innovations in personalized LLMs.

## 3 Foundations of Personalized LLMs

While previous research (Yang & Flek, 2021; Chen et al., 2024c;b) has explored definitions and analyzed various aspects of personalized LLMs, a comprehensive theoretical framework for understanding and formalizing personalization in these models is still lacking. In this section, we aim to fill this gap by establishing the foundational principles, definitions, and formal structures to formalize the problem of personalization in LLMs. We systematically develop the necessary notation and conceptual framework to formalize the problem and evaluation, setting the stage for a deeper understanding of how personalization can be effectively implemented and analyzed within LLMs. The following subsections are structured as follows:

**§3.1 General Principles of LLMs:** We begin by outlining the core principles that form the foundation of LLMs. This provides essential context for understanding how these models function and the underlying mechanics that drive their capabilities.

**§3.2 Definition of Personalization in LLMs:** We define the term "personalization" within the specific context of LLMs, establishing a clear understanding for subsequent discussions.

**§3.3 Overview of Personalization Data:** We provide an overview of the current data utilized for personalization, emphasizing the different formats of data sources.

**§3.4 Formalization of Personalized Generation:** We formalize the conceptual space for personalized generation, providing a structured framework for understanding how personalization can be achieved.

**§3.5 Taxonomy of Personalization Criteria:** We introduce a comprehensive taxonomy of personalization criteria, categorizing the various factors that influence personalized outputs.

### 3.1 Preliminaries

Let $\mathcal{M}$ be an LLM parameterized by $\theta$, which takes a text sequence $X \in \mathbb{X}$ as input and produces an output sequence $\hat{Y} \in \hat{\mathbb{Y}}$, where $\hat{Y} = \mathcal{M}(X; \theta)$. The form of $\hat{Y}$ depends on the specific task, with $\hat{\mathbb{Y}}$ representing the output space of possible generations. The inputs can be drawn from a labeled dataset $\mathcal{D} = (X^{(1)}, Y^{(1)}), \cdots, (X^{(N)}, Y^{(N)})$, or from an unlabeled dataset of prompts for sentence continuations or completions $\mathcal{D} = X^{(1)}, \cdots, X^{(N)}$. For this and other notation, see Table 2.

**Definition 1** (LARGE LANGUAGE MODEL). *A large language model (LLM) $\mathcal{M}$, parameterized by $\theta$, is a multi-layer Transformer model with billions (or more) of parameters. It can be structured with an encoder-only, decoder-only, or encoder-decoder architecture and is trained on extensive corpora comprising a vast number of natural language tokens (Zhao et al., 2023; Gallegos et al., 2024).*

**Definition 2** (DOWNSTREAM TASKS). *A downstream task is a specific practical application or goal, such as classification, translation, recommendation, or information retrieval, that uses outputs generated by a model (e.g., an LLM). Formally, for a downstream task, we define a corresponding downstream model or function $\mathcal{F}$ that takes as input the model's output $\hat{y}$ (generated from an initial input $X$) and produces a final result or prediction*

$$\hat{r} = \mathcal{F}(\hat{y})$$

Currently, LLMs are mainly built upon multi-layer Transformer (Vaswani et al., 2017), which employ stacked multi-head attention layers within a deeply structured neural network (Zhao et al., 2023). Based on the use of different components of the original transformer architecture, LLMs can be categorized into the following three categories: (1) decoder-only models (e.g., GPT series (Radford et al., 2018; 2019; Brown et al., 2020; Achiam et al., 2023)) (2) encoder-only models (e.g., BERT-based models (Devlin et al., 2018; Liu et al.,

2019)), (3) encoder-decoder models (e.g., T5 (Raffel et al., 2020)). Among those categories, decoder-only LLMs become the most popular type which is optimized for next-token generations.

After pre-training with large-scale unlabeled corpora in an unsupervised manner, the resulting in-context-aware word representations are very effective as general-purpose semantic features for a wide range of NLP tasks. With the scaling of their size and techniques such as instruction tuning (Ouyang et al., 2022; Zhang et al., 2023c; Longpre et al., 2023; Zhou et al., 2024a) and RLHF (Christiano et al., 2017; Stiennon et al., 2020b; Rafailov et al., 2024), LLMs exhibit many emergent abilities (Wei et al., 2022a). This enables LLMs to solve complex tasks and engage in natural conversations with humans, even in a zero-shot manner through text prompting for a wide range of downstream tasks such as sequence classification, text generation, and recommendation (Qin et al., 2023). To further enhance LLMs' performance on specific downstream tasks, models are often fine-tuned with a relatively small amount of task-specific data following the "pre-train, then fine-tune" paradigm, which generally adapts LLMs to particular tasks and achieves better results (Bommasani et al., 2021; Min et al., 2023; Liu et al., 2023b).

**Definition 3** (PROMPT). *A Prompt $\mathcal{H}$ is a specific input or set of instructions provided to a language model, which guides its generation of text $M(X; \theta)$. Prompts can vary in complexity from simple word or phrase completions to detailed, structured contexts or questions aimed at eliciting specific types of responses or performing certain tasks. Prompts can be multi-modal, including text, image, audio, or video inputs.*

- ***System Prompt:*** *A* System Prompt $\mathcal{H}_{sys}$ *is a predefined prompt that initializes the interaction, setting the overall behavior, style, or constraints of the language model. It often provides consistent instructions on how the model should respond to subsequent user prompts throughout the interaction. This is particularly useful for role-playing or establishing the model's tone and persona.*

- ***User Prompt:*** *A* User Prompt $\mathcal{H}_{usr}$ *is an input provided by the user during the interaction with the language model, typically seeking specific information, responses, or actions from the model. For simplicity, in the following sections, we will represent the user prompt as $x$.*

### 3.2 Formulation of Personalization

**Definition 4** (PERSONALIZATION). Personalization *refers to the process of tailoring a system's output to meet the individual preferences, needs, and characteristics of an individual user or a group of users. In the context of LLMs, personalization involves adjusting the model's responses based on user-specific data, historical interactions, and contextual information to enhance user satisfaction and relevance of the entire system's generated content.*

**Definition 5** (USER PREFERENCES). User Preferences *refer to the specific likes, dislikes, interests, and priorities of an individual user or a group of users. These preferences guide the personalization process by informing the system about the desired characteristics and features of the output. In the context of LLMs, user preferences can be derived from explicit feedback (e.g., pairwise comparison), historical interactions, and contextual signals to tailor responses and improve the relevance and satisfaction of the generated content.*

**Definition 6** (PERSONALIZED LARGE LANGUAGE MODEL). *A* Personalized Large Language Model *(Personalized LLM) $\mathcal{M}_p$ is an LLM that has been adapted to align with the individual preferences, needs, and characteristics of a specific user or group of users. This adaptation involves utilizing user-specific data, historical interactions, and contextual information to modify the model's responses, making them more relevant and satisfying for the user. Personalized LLMs aim to enhance the user experience by providing tailored content that meets the unique expectations and requirements of the user.*

**Definition 7** (USER DOCUMENTS). User Documents $\mathcal{D}_u$ *refer to the collection of texts and writings generated by a user $u$. This includes reviews, comments, social media posts, and other forms of written content that provide insights into the user's preferences, opinions, and sentiments.*

**Definition 8** (USER ATTRIBUTES). User Attributes $A_u = \{a_1, a_2, \ldots, a_k\}$ *are the static characteristics and demographic information associated with a user $u \in U$. These attributes include age, gender, location, occupation, and other metadata that remain relatively constant over time.*

**Definition 9** (USER INTERACTIONS). User Interactions $I_u = \{i_1, i_2, \ldots, i_m\}$ *capture the dynamic behaviors and activities of a user $u \in U$ within a system. This includes actions such as clicks, views, purchases, and other engagement data that reflect the user's preferences and interests.*

Personalization, the practice of tailoring experiences to the preferences of individual users or groups of users, is crucial for bridging the gap between humans and machines (Rossi et al., 1996; Montgomery et al., 2004; Chen et al., 2024c). Such experiences can include aligning with specific user or group preferences, adjusting the style or tone of generated content, and providing recommendation items based on the user's interaction history in a wide range of downstream tasks. Users can be actual individuals with a history of interactions or described by specific characteristics such as demographic information, allowing both humans and machines to better understand and cater to their needs. In this work, instead of just focusing on personalization for single individual users, we aim to formalize and clarify the term "personalization" by categorizing its objectives based on the size of the targeted group. We classify personalization into three categories based on their focus: aligning with the preferences of individual users, groups of users, or the general public (Sec. 4). Additionally, these three levels of personalization enable the incorporation of different types of input data, each contributing uniquely to the personalization process. It is important to note that not all fine-tuning equates to personalization. For example, most supervised fine-tuning practice is a process where models are trained on specific datasets to perform better on a downstream task. However, only fine-tuning that adjusts a model to cater to specific user or group preferences—such as adapting a model to a user's writing style or content preferences—counts as personalization. In contrast, fine-tuning on a general corpus to improve overall task performance is not personalized, as it does not address the unique preferences of individuals or groups. This distinction is key to understanding the objectives of personalized LLMs across the different levels of granularity.

## 3.3 Personalization Data

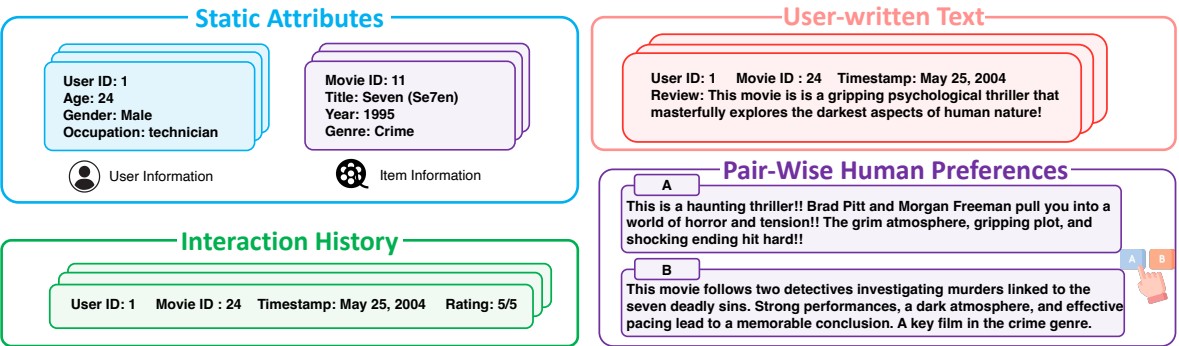

Figure 2: **Overview of Personalization Data.** This figure presents an overview of the various types of user-specific data used in downstream personalization tasks. It categorizes the data into three primary formats: (i) **Static Attributes**, which include demographic information and item metadata that remain relatively constant over time; (ii) **Interaction History**, capturing dynamic user behaviors and preferences through previous activities and engagement data; (iii) **User-Written Text**, encompassing reviews, dialogues, and social media posts that provide rich insights into user sentiment and preferences; and (iv) **Pair-Wise Human Preferences**, explicit feedback or annotations that guide the system to align with individual user needs.

In this section, we provide an overview of various formats of user-specific information commonly used in downstream personalization tasks. Understanding such data is critical for leveraging user information and designing targeted personalization techniques to enhance the performance of LLMs in diverse applications. Figure 2 illustrates this overview with concrete examples.

### 3.3.1 Static Attributes

Static attributes refer to information about both users and items that remain relatively constant over time. These attributes form the foundation of many personalization strategies and are often used to segment users and items for more targeted recommendations. Except for unique identifiers assigned to each user and item, such as User ID and Item ID, common static attributes include:

- **User's Demographic Information**: Age, gender, location, and occupation can help infer preferences and tailor content or product recommendations.

- **Item Information**: For recommendation systems, item-specific data, such as title, release date, genre, and other relevant metadata, play a crucial role in understanding user preferences and making accurate recommendations.

Static attributes provide a reliable basis for long-term personalization strategies. Typically collected during user registration or profile setup for users, and during the cataloging process for items, this data requires minimal human effort for annotation. However, static attributes do not capture changes in user preferences or item relevance over time, which limits their effectiveness in downstream personalization tasks. Additionally, collecting and storing demographic information can raise privacy issues, necessitating careful handling and compliance with data protection regulations. Techniques for anonymizing data (Samarati & Sweeney, 1998) are essential to address these concerns.

### 3.3.2 Interaction History

Interaction history captures the dynamic aspects of a user's behavior and preferences based on their interactions with a system. This data is crucial for understanding user preferences and enabling real-time personalized recommendations. Interaction history includes information about past activities, such as movies watched, songs listened to, items purchased, or articles read. It also covers user interactions with items they have clicked on or viewed, including engagement duration, which helps infer interests and engagement levels. Additionally, in the context of interactions with LLMs, this history includes the content of previous prompts, responses, and the patterns of user engagement with the generated outputs, all of which contribute to tailoring future interactions.

The advantage of interaction history is its dynamic and up-to-date nature, providing real-time insights into user preferences and enabling timely and relevant recommendations. Detailed interaction data offers rich context, aiding in a deeper understanding of user behavior. However, interaction history can be voluminous and complex to process, requiring sophisticated data-handling techniques. Additionally, past interactions may not always accurately reflect current preferences, necessitating careful analysis to maintain relevance.

### 3.3.3 User-Written Text

User-written text includes any form of written content generated by users, such as reviews, comments, dialogues, or social media posts. This type of data is rich in user sentiment and can provide deep insights into user preferences and opinions. User text data typically encompasses:

- **Reviews**: Written evaluations of products or services, often including ratings and detailed comments. For example, the Amazon Review Data (Ni et al., 2019) contains 233.1 million reviews, offering insights into user experiences and preferences through detailed textual feedback and ratings.

- **Dialogues and Conversations**: Textual exchanges between users and dialogue systems or other users. The ConvAI2 (Dinan et al., 2020) dataset includes dialogues where participants are assigned personas and engage in natural conversations, helping to understand user interaction patterns and improve conversational agents.

- **Social Media Posts**: Short messages or comments on platforms like Reddit, Twitter, or Facebook, which can be analyzed to understand user sentiments and trends.

In the context of LLMs, this also includes human-written exemplars often used for few-shot learning, reflecting user preferences or intent to guide the model's responses. The potential use cases for user text data are extensive. For example, sentiment analysis (Medhat et al., 2014; Wankhade et al., 2022) can be performed to understand user opinions and improve product offerings or customer service. Conversational agents can be enhanced by analyzing user conversations to make interactions more natural and engaging. The advantages of user text data lie in its depth of insight, providing detailed information about user preferences, opinions, and sentiments. It is versatile and applicable across various domains, from product reviews to social media analysis. However, text data is inherently unstructured, necessitating advanced NLP techniques for effective

analysis. Besides, comprehensively evaluating such nuanced data, especially for personalization, is challenging with existing metrics. Additionally, user-generated content can be noisy and variable in quality, complicating accurate analysis. Annotating high-quality new data points is expensive, further adding to the complexity.

### 3.3.4 Pair-Wise Human Preferences

Pair-wise human preferences refer to explicit user feedback indicating their preferred responses from a set of candidate outputs. This data format typically involves human annotations selecting the most desired option, making it essential for training models to align closely with individual user needs and preferences. Unlike static attributes or interaction history, pair-wise preferences offer highly specific and direct feedback, serving as explicit instructions on how users expect the model to behave or respond in given scenarios. For example, users might specify whether they want a response to be easily understood by a layperson or tailored for an expert. In this way, users can explicitly state what they want, reducing ambiguity and implicitly, which can be useful leading to higher user satisfaction and more effective personalization. However, designing an appropriate alignment strategy remains a significant challenge for personalization applications. Most current works focus on aligning models with general, aggregate human preferences, rather than diverse, individual perspectives (Jang et al., 2023). Developing methods to capture and use these individual direct preferences effectively is essential for advancing personalized systems.

**Definition 10** (ALIGNMENT). *Alignment $\mathcal{G}$ is the process or state by which an AI system's goals, $\mathcal{G}_A$, is consistent with human values and intentions, denoted as $\mathcal{G}_H$. Mathematically, alignment can be defined as ensuring that the behavior policy $\pi_A$ of the AI system maximizes the utility function $U_H$ representing human values. Formally,*

$$\mathcal{G} = \{\pi_A \mid \pi_A \in \arg\max_\pi \mathbb{E}_\pi [U_H]\}$$

*where $\pi_A$ is the policy of the AI system, $\mathbb{E}_\pi [U_H]$ is the expected utility under policy $\pi$, and $\arg\max_\pi$ denotes the set of policies that maximize the expected human utility $U_H$.*

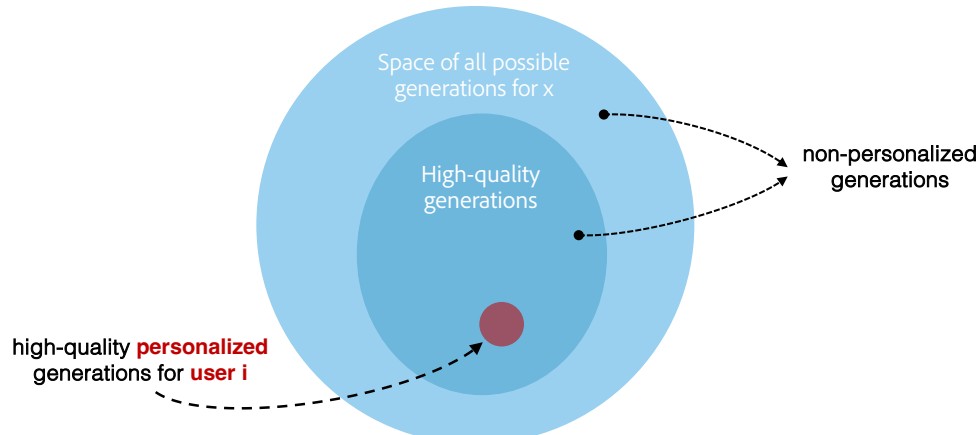

Figure 3: **Space of Personalized Generations.** We characterize the space of generations for query $x$, including the space of all possible generations $\mathcal{S}(x)$, the space of all high quality generations $\mathcal{S}_h(x)$, and finally, the space of user-specific high-quality personalized generations $\mathcal{S}_i(x)$ for user $i$. Intuitively, given two users $i$ and $j$, the space of high-quality personalized generations for each user may be completely disjoint.

### 3.4 Space of Personalized Generations

In this section, we briefly formalize and analyze the problem of personalized LLMs and their solution spaces and provide an intuitive overview in Figure 3. This serves two purposes: providing intuition on the difficulty of the problem and characterizing the properties and unique advantages that relate to other well-studied problems.

Let us first establish the formalization of the personalized LLM problem. Consider a generic input example $x \in \mathcal{X}$. We denote the generative model by $g : \mathcal{Z} \times \mathcal{X} \to \mathcal{Y}$, where $\mathcal{Z}$ represents the latent space and $\mathcal{Y}$ represents the space of all possible generations. Given input $x \in \mathcal{X}$, the space of all possible generations is

$$\mathcal{S}(x) = \{g(\mathbf{z}, x) : \mathbf{z} \in \mathcal{Z}\} \subseteq \mathcal{Y}$$

To facilitate a comprehensive understanding of personalization, we delineate the following sets:

- The space of all possible generations $\mathcal{Y}$.
- The space of all possible generations for a given input $x$ is $\mathcal{S}(x)$.
- The space of high-probability generations for a given input $x$, denoted by:

$$\mathcal{S}_h(x) = \{y \in \mathcal{Y} : P(y|x) \geq \delta\}$$

  where $P(y|x)$ is the probability of generation $y$ given input $x$ and $\delta$ is a threshold representing high-quality content.

- The space of user-specific generations for a user $u_i \in U$ given input $x$, denoted by:

$$\mathcal{S}_i(x) = \{y \in \mathcal{S}_h(x) : f(P_{u_i}, y) \leq \epsilon, P(y|x) \geq \delta\}$$

  where $f(P_{u_i}, y)$ is a function that quantifies the alignment of the generation $y$ with the user's preferences $P_{u_i}$, and $\epsilon$ is a threshold for user-specific relevance.

An intuitive overview of the space of personalized generations for a specific user can be found in Figure 3. Note that the space of user-specific generations $\mathcal{S}_i(x)$ is significantly smaller and more targeted compared to the space of all possible generations $\mathcal{S}(x)$. One example is that there may be many correct responses to a specific question, however, only a very small subset of answers may capture the important details needed for the answer to be useful to the specific user. In particular, the user may need additional steps to carry out a specific task, or they may know certain terminology and language that would enable better understanding for the user. Overall, the key takeaways are that the user personalization tasks are extremely challenging, though will only become increasingly important in the future. In particular, there is only a very tiny space of responses that are useful to a specific user, and generating such a response is only becoming more and more important. This highlights the need to not only develop better techniques to generate such user-specific responses, but also better data and evaluation metrics to quantify it.

### 3.5 Personalization Criterion Taxonomy

When evaluating the personalization of generated text in LLMs, it is essential to consider several critical aspects to ensure the content is effectively tailored to individual users. These aspects constitute a taxonomy of personalization criteria, encompassing various dimensions of personalized content generation.

**Tone and Style**   One of the fundamental aspects of personalized text generation is the alignment of tone and style with the user's preferences and previous interactions. This includes:

- **Writing Style**: The writing style should be consistent with the user's preferred style or previous interactions. For instance, if a user typically prefers a more concise style for an email, the generated text should reflect such preference, ensuring a seamless user experience.

- **Tone**: The tone of the generated content should match the user's preferred tone, which could vary depending on the context. For example, the tone could be formal, casual, professional, or friendly, depending on the user's past written texts and the situational requirements.

**Relevance**   Personalization also necessitates that the generated content be highly relevant to the user's interests, preferences, and current needs. This relevance is assessed on two levels:

- **Content Relevance**: This criterion evaluates whether the content aligns with the user's interests and preferences. It ensures that the generated text is pertinent and valuable to the user, thus enhancing

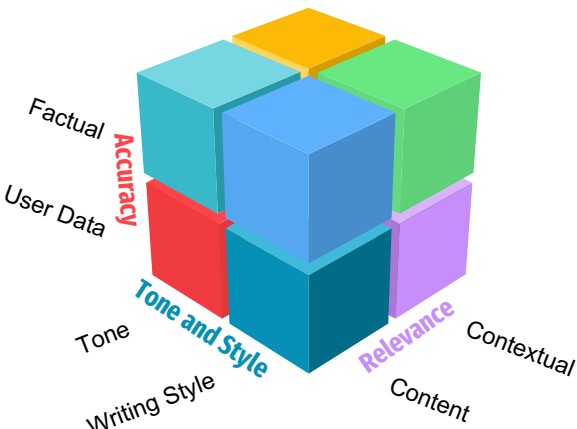

Figure 4: **Dimensions in Personalized Criterion.** We propose a framework that expands the dimensions of personalization criteria LLMs along three aspects: (i) Tone and Style, which includes writing style and tone preferences to match the user-written text; (ii) Relevance, encompassing content relevance to user interests and contextual relevance for specific situations; and (iii) Accuracy, which ensures both factual correctness and accurate representation of user data. These aspects interact to form a comprehensive taxonomy, addressing the multi-faceted nature of effective personalization in LLM-generated text.

engagement and satisfaction. For example, if a user has recently shown interest in sustainability topics, the LLM should prioritize generating content related to green technologies or eco-friendly practices in relevant contexts, such as when drafting blog posts or social media updates.

- **Contextual Relevance**: Beyond general interests, it is crucial that the content is appropriate for the specific context or situation in which the user will encounter it. For example, if the user is preparing for a business presentation, the LLM should focus on generating content that is formal, data-driven, and aligned with the specific industry, rather than casual or unrelated topics.

**Accuracy** Accuracy is another critical dimension of personalized text generation, ensuring that the information provided is reliable and precise. This includes:

- **Factual Accuracy**: The generated content should be factually correct and based on reliable information. This ensures the credibility of the content and maintains the trust of the user. For example, if the LLM is generating a report on recent market trends, it should use up-to-date data and cite reliable sources, avoiding any outdated or incorrect information.

- **User Data Accuracy**: Personalization heavily depends on the accuracy of the user data used to tailor the content. The personalized content must be based on up-to-date and correct user data, which includes the user's preferences, past behavior, and interactions. For example, if a user recently changed their job title from 'Manager' to 'Director,' the LLM should generate emails or documents that reflect this new role and associated responsibilities, rather than using outdated information.

These aspects of personalization—tone and style, relevance, and accuracy—form the foundation of a robust taxonomy for evaluating personalized LLMs. Each criterion plays a vital role in ensuring that the generated content is tailored effectively, providing a unique and satisfying user experience. This taxonomy not only aids in the systematic evaluation of personalized LLMs but also highlights the multi-faceted nature of personalization. By addressing each of these criteria, researchers and practitioners can develop more sophisticated and user-centric language models that better serve the diverse needs and preferences of users.

Table 1 provides an illustrative breakdown of these criteria, along with their respective descriptions and examples.

Table 1: **Taxonomy of Personalized LLM Criterion.**

| Criterion | Description and Examples |
|---|---|
| TONE AND STYLE | |
| **Writing Style** | Is the writing style consistent with the user's preferred style or previous interactions? |
| **Tone** | Does the tone of the text match the user's preferences (previous written text) and context (*e.g.*, formal, casual, etc)? |
| RELEVANCE | |
| **Content Relevance** | Does the content match the user's interests, preferences, and needs? |
| **Contextual Relevance** | Is the content appropriate for the specific context/situation that the user will encounter it? |
| ACCURACY | |
| **Factual Accuracy** | Are the facts and information presented in the text correct and reliable? |
| **User Data Accuracy** | Is the personalized content based on accurate and up-to-date user data? |

## 3.6 Overview of Taxonomies

In this section, we present a high-level summary of each taxonomy proposed in the subsequent sections of the paper. Comprehensive descriptions of these taxonomies can be found in Sections 4, 5, 6, and 7.

### 3.6.1 Taxonomy of Personalization Granularity of LLMs

We propose three different levels of personalization granularity for LLMs, each addressing different scopes of personalization. These levels help in understanding the depth and breadth of personalization that can be achieved with LLMs. The three levels are:

**§4.1 User-level Personalization:** Focuses on the unique preferences and behaviors of a single user. Personalization at this level utilizes detailed information about the user, including their historical interactions, preferences, and behaviors, often identified through a user ID.

**§4.2 Persona-level Personalization:** Targets groups of users who share similar characteristics or preferences, known as personas. Personalization here is based on the collective attributes of these groups, such as expertise, informativeness, and style preferences.

**§4.3 Global Preference Personalization:** Encompasses general preferences and norms that are widely accepted by the general public, such as cultural standards and social norms.

### 3.6.2 Taxonomy of Personalization Techniques for LLMs

We categorize personalization techniques for LLMs based on the way user information is utilized. These techniques provide various methods to incorporate user-specific data into LLMs to achieve personalization. The main categories are:

**§5.1 Personalization via Retrieval-Augmented Generation:** Incorporates user information as an external knowledge base, encoded through vectors, and retrieves relevant information using embedding space similarity search for downstream personalization tasks.

**§5.2 Personalization via Prompting:** Incorporates user information as the context within the prompts for LLMs, allowing for downstream personalization tasks.

**§5.3 Personalization via Representation Learning:** Encodes user information into the embedding spaces of neural network modules, which can be represented through model parameters or explicit embedding vectors specific to each user.

**§5.4 Personalization via Reinforcement Learning From Human Feedback:** Uses user information as the reward signal to align LLMs with personalized preferences through reinforcement learning.

### 3.6.3 Taxonomy of Evaluation Methodologies for Personalized LLMs

Evaluation metrics for personalized LLMs can be classified based on how they measure the effectiveness of personalization. These metrics ensure that the personalized outputs meet the desired standards of relevance and quality. The main categories are:

**§6.1 Intrinsic Evaluation:** Evaluates the personalized text generated directly, focusing on factors like personalized content, writing style, and more.

**§6.2 Extrinsic Evaluation:** Relies on downstream applications such as recommendation systems to demonstrate the utility of the generated text from the personalized LLM.

### 3.6.4 Taxonomy of Datasets for Personalized LLMs

We propose a taxonomy that categorizes personalized LLM datasets based on whether they contain text written by specific users. This helps in understanding the data's role in training or evaluating personalized LLMs directly or indirectly. The main categories are:

**§7.1 Personalized Datasets *with* Ground-Truth Text:** Contain actual ground-truth text written by users, enabling direct evaluation of personalized text generation.

**§7.2 Personalized Datasets *without* Ground-Truth Text:** Used for indirect evaluation via downstream applications, as they do not contain user-specific ground-truth text.

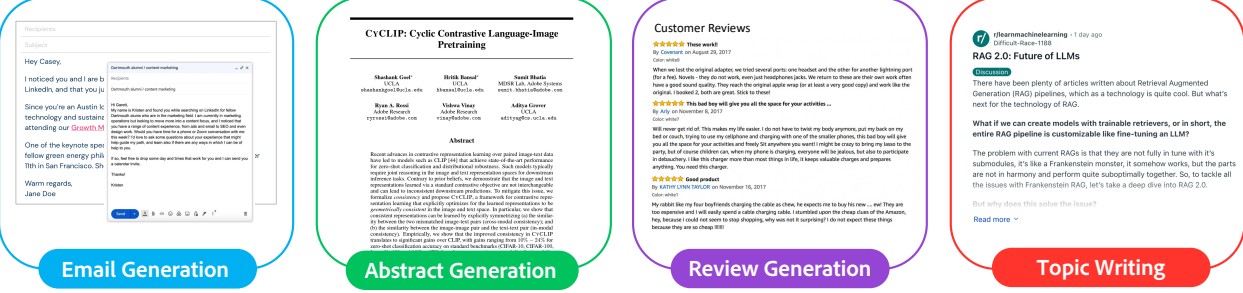

Figure 5: **Examples of Personalization Tasks and Data.**

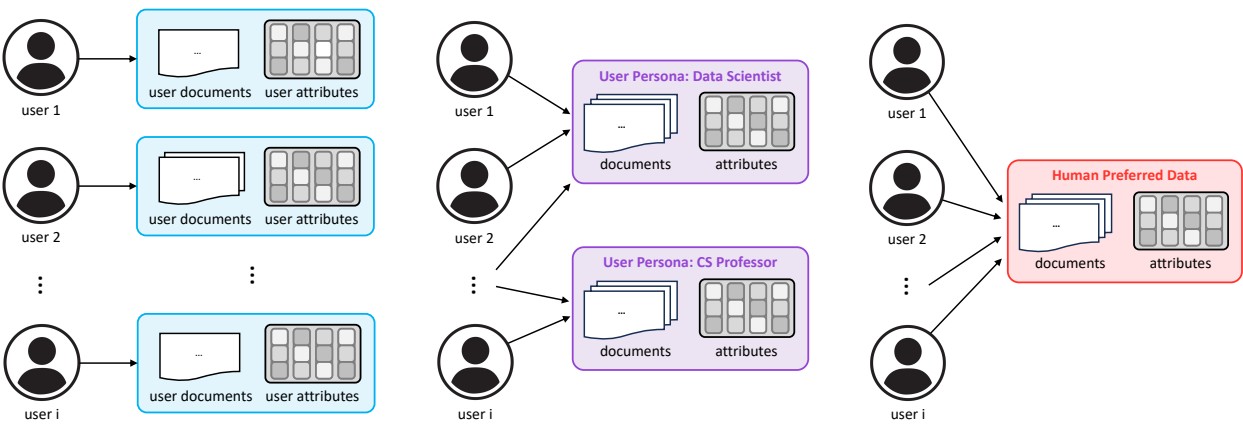

(a) User-level Personalization (§4.1)   (b) Persona-level Personalization (§4.2)   (c) Global Preference Personalization (§4.3)

Figure 6: **Personalization Granularity Taxonomy.**

Table 2: **Summary of key notation.**

| Notation | Definition |
|---|---|
| $\mathcal{D}$ | dataset |
| $\mathcal{D}_i$ | user $i$'s specific user data |
| $t_i$ | text written by user $i$ |
| $a_i$ | attributes/preferences of user $i$ |
| $I_i$ | interactions of user $i$ |
| $X_i = (x_1, \cdots, x_m) \in \mathbb{X}$ | generic input text for user $i$ |
| $\bar{x}$ | transformed personalized input based on retrieved user information |
| $Y_i \in \mathbb{Y}$ | ground-truth text for user $i$ |
| $\hat{Y}_i \in \hat{\mathbb{Y}}$ | personalized text generated for user $i$ |
| $\mathbf{v}$ | task-specific feature vector |
| $U$ | A set of users |
| $i$ | A single user $i \in U$ |
| $\mathcal{S}$ | set of personas |
| $S$ | a persona $S \in \mathcal{S}$ |
| $P_u$ | The set of a single user's preferences |
| $P_s$ | The set of a persona $s$'s preferences |
| $P_G$ | Global preferences |
| $\mathbf{r}$ | Downstream tasks' label |
| $\hat{\mathbf{r}}$ | Model's predictions on downstream task |
| $\mathcal{G}$ | the process of alignment |
| $\mathcal{G}_\mathcal{A}$ | AI system's targeted preferences |
| $\mathcal{G}_\mathcal{H}$ | human's values and intentions |
| $\pi_A$ | AI system's behavior policy |
| $U_H$ | the utility function which represents human values |
| $\mathbb{E}_i$ | intrinsic evaluation |
| $\mathbb{E}_e$ | extrinsic evaluation |
| $\psi(\cdot) \in \Psi$ | an evaluation metric |
| $\psi_a(\cdot) \in \Psi_a$ | an evaluation metric for downstream application |
| $\mathcal{L}(\cdot)$ | loss function |
| $\mathcal{M}$ | LLM |
| $\mathcal{M}_p$ | personalized LLM |
| $\mathcal{H}_{sys}$ | The system prompt to input LLMs |
| $\mathcal{H}_{usr}$ | The user prompt to input LLMs |
| $g$ | generative model |
| $E(\cdot)$ | word or sentence encoder, which can be a part of $\mathcal{M}$ |
| $\mathcal{R}(\cdot)$ | retrieval model |
| $RecSys$ | recommendation system |
| $\mathcal{F}(\cdot)$ | downstream model such as recommendation system |
| $\phi_q$ | query construction function |
| $\phi_p$ | personalized prompt construction function |
| $\mathbf{z}$ | embedding of generated text $\hat{Y}_i$ for user $i$ |
| $\mathbf{r}$ | the output of a personalized system for a downstream task |

# 4 Personalization Granularity of LLMs

**Definition 11** (PERSONALIZATION GRANULARITY)**.** Personalization Granularity *refers to the level of detail at which personalization objectives are defined and implemented. It determines the extent to which the system's responses are tailored to specific criteria, such as individual users, groups of users with a certain shared persona, or the general public, influencing how finely or broadly the personalization is applied.*

In this section, we propose a taxonomy for personalized LLMs based on the granularity of the personalization objective. Specifically, personalized LLMs can be categorized by their focus on aligning with the preferences of individual users, groups of users, or the general public. In this survey, we formally define the granularity of personalization with the following distinctions:

- **User-level Personalization (Sec. 4.1)**: This level focuses on the unique preferences and behaviors of a single user. Personalization at this level utilizes detailed information about the user, including their historical interactions, preferences, and behaviors, often identified through a user ID (Li et al., 2024g). Formally, let $U$ represent the set of users, and $P_u = \{p_u^1, p_u^2, ..., p_u^n\}$ denote the set of personalized preferences for user $u \in U$. The objective function of the downstream task is $\mathcal{L}_{task}$. The objective of personalization on this level is to minimize this function:

$$\theta^* = \operatorname*{argmin}_{\theta} \mathcal{L}_{task}(f_\theta(P_u))$$

  where $\theta$ can be parameters or prompts in the LLM-based system $f$.

- **Persona-level Personalization (Sec. 4.2)**: This level targets groups of users who share similar characteristics or preferences, known as personas. Personalization here is based on the collective attributes of these groups, such as expertise, informativeness, and style preferences (Jang et al., 2023). Formally, let $S$ represent the set of personas, where each persona $s \in S$ is composed of a subset of users $U_s \subseteq U$ with shared characteristics or preferences. Let $P_g$ denote the set of personalized preferences for persona $s$. For every preference $p_i \in P_s$ and every user $u \in U_s$, it holds that $p_i \in P_u$. The objective of personalization on this level is to minimize this function:

$$\theta^* = \operatorname*{argmin}_{\theta} \mathcal{L}_{task}(f_\theta(P_s))$$

- **Global Preference Personalization (Sec. 4.3)**: This level encompasses general preferences and norms that are widely accepted by the general public. For example, broadly accepted cultural standards and social norms. Formally, let $P_{global}$ represent the set of universal preferences. For every preference $p_i \in P_{global}$ and every user $u \in U$, it holds that $p_i \in P_u$. The objective of personalization on this level is to minimize this function:

$$\theta^* = \operatorname*{argmin}_{\theta} \mathcal{L}_{task}(f_\theta(P_{global}))$$

## 4.1 User-level Personalization

In this section, we discuss user-level personalization, which focuses on data at the individual level (Zollo et al., 2024). As depicted in Figure 6(a), this type of personalization focuses on optimizing preferences for each user uniquely identified by a user ID. For instance, in the MovieLens-1M recommendation dataset (Harper & Konstan, 2015), each user has demographic information such as *UserID*, *Gender*, *Age*, *Occupation*, and *Zip-code*, alongside corresponding movie interactions (MovieID, Rating, Timestamp). The goal is to recommend new movies based on each user's profile and viewing history. The advantage of this level of personalization is that it offers the most fine-grained approach, minimizing noise from other users. This is particularly beneficial in domains such as online shopping, job recommendations (Wu et al., 2024b), and healthcare (Abbasian et al., 2023; 2024; Zhang et al., 2024a; Jin et al., 2024b), where individual user behavior can vary significantly, and such detailed, individualized personalization is crucial. One of the main challenges of this level of personalization is the "cold-start problem" which refers to users with minimal interaction history, often termed "lurkers" in recommendation systems (Sun et al., 2024). However, many studies (Salemi et al., 2023; Rajput et al., 2023; Xi et al., 2023) choose to remove such data during the preprocessing stages. This exclusion potentially undermines the robustness of the systems by disregarding the subtleties and potential insights offered by these underrepresented user interactions.

## 4.2 Persona-level Personalization

In this section, we discuss persona-level personalization, where the input comprises the preferences of users categorized by group or persona. As illustrated in Figure 6(b), this approach targets optimizing the preferences of a user group sharing common characteristics. A natural language description encapsulating these shared traits represents the entire group within prompts or other relevant components. For example, Jang et al. (2023) design three distinct perspectives of preferences: expertise, informativeness, and style, with each dimension featuring two conflicting personas or preferences. For instance, in the expertise dimension, one

persona prefers content that is easily understandable by an elementary school student, while the other prefers content that is comparable only to a PhD student in the specific field. From this example, we can observe that, compared to localized user-specific personalization (Sec. 4.1), each persona represents a broader portrait of a group of users, focusing on more general features rather than detailed user-specific information. The advantage of persona-level personalization lies in its effectiveness in scenarios where shared characteristics are prominent and crucial for downstream tasks, while user-specific attributes are less significant. Additionally, once these characteristics are extracted, this data format is easier to process, either by including it directly in the prompt or utilizing it through RLHF, compared to lengthy user-specific profiles. However, extracting representative characteristics through natural language descriptions can be challenging in practice, often requiring substantial reliance on human domain knowledge.

### 4.3 Global Preference Personalization

In many applications, only global user preference data may be available, representing the preferences of the entire population rather than those of individual users. While this falls outside the primary scope of personalization in this survey, we include a discussion of it for completeness. These preferences typically encompass human values expected to be accepted by the general public, such as social norms, factual correctness, and instruction following (Taylor et al., 2016; Gabriel, 2020; Liu, 2024). The common format of such data includes a given instruction, multiple options, and a label annotated by human annotators indicating which option is preferable (Ethayarajh et al., 2022; Stiennon et al., 2020a; Nakano et al., 2021; Bai et al., 2022; Ganguli et al., 2022). These datasets are typically used through RLHF to align LLMs. The advantage of global preference alignment is its potential to enhance LLMs in terms of safety (Gehman et al., 2020; Ge et al., 2023; Anwar et al., 2024; Ji et al., 2024a), social norms (Ryan et al., 2024), and ethical issues (Liu et al., 2021; Rao et al., 2023), ensuring they align with human values. However, the disadvantage is that it may introduce noise, as individual preferences can vary and may not always represent the general public accurately. Moreover, this level of alignment does not capture fine-grained personalization.

### 4.4 Discussion

The granularity of personalization in LLMs involves trade-offs between *precision*, *scalability*, and *richness* of personalized experiences. User-level personalization offers high precision and engagement but faces challenges with data sparsity and scalability. Persona-level personalization is efficient and representative but less granular and requires domain knowledge for defining personas. Global preference personalization provides broad applicability and simplicity but lacks specificity and can introduce noise from aggregated data. In the future, hybrid approaches may leverage the strengths of each method while mitigating their weaknesses. For instance, a hierarchical personalization framework can combine user-level personalization for frequent users, persona-level personalization for occasional users, and global preferences for new users. This balances precision and scalability by tailoring experiences based on user interaction levels. Another idea is *context-aware personalization*, which starts with persona-level personalization and transitions to user-level as more data becomes available, addressing the cold-start problem. This approach allows the system to offer relevant personalization initially and gradually refine it with detailed user-specific data. Such adaptive systems can dynamically adjust the granularity based on user engagement, context, and data availability. These systems can switch between levels of personalization, providing a balanced and effective user experience by utilizing the most appropriate granularity for each situation. Integrating information across different granularities may further enhance personalization. User-level data can refine persona definitions, making them more accurate and representative. Conversely, persona-level insights can inform user-level personalization by providing context on shared characteristics. Global preferences can serve as a baseline, ensuring that individual and persona-level personalization aligns with broadly accepted norms and values. Currently, datasets for these three levels of granularity are often orthogonal and unrelated. Developing datasets that encompass user-level, persona-level, and global preferences is crucial. Such datasets would enable more seamless integration and transition between different levels of personalization, enhancing the robustness and effectiveness of LLMs in catering to diverse user needs. In conclusion, the choice of personalization granularity should be guided by specific application requirements, balancing precision, scalability, and the ability to

provide rich, personalized experiences. Hybrid approaches and integrated datasets are key to achieving optimal personalization outcomes.

Table 3: **Taxonomy of Techniques for Personalized LLMs.**

| Category | Mechanism |
|---|---|
| PERSONALIZATION VIA RAG (§ 5.1) | Sparse Retrieval (§ 5.1.1) |
| | Dense Retrieval (§ 5.1.2) |
| PERSONALIZATION VIA PROMPTING (§ 5.2) | Contextual Prompting (§ 5.2.1) |
| | Persona-based Prompting (§ 5.2.2) |
| | Profile-Augmented Prompting (§ 5.2.3) |
| | Prompt Refinement (§ 5.2.4) |
| PERSONALIZATION VIA REPRESENTATION LEARNING (§ 5.3) | Full-Parameter Fine-tuning (§ 5.3.1) |
| | Parameter-Efficient Fine-tuning (§ 5.3.2) |
| | Embedding Learning (§ 5.3.3) |
| PERSONALIZATION VIA RLHF  (§ 5.4) | |

## 5 Taxonomy of Personalization Techniques for LLMs

In this section, we propose a taxonomy of personalization techniques for LLMs categorized by the way user information is utilized. In particular, techniques for personalizing LLMs are categorized as follows:

- **Personalization via RAG (Sec 5.1)**: This category of methods incorporates user information as an external knowledge base, encoded through vectors. When new inference data arrives, the relevant information is retrieved using embedding space similarity search for downstream personalization tasks.

- **Personalization via Prompting (Sec 5.2)**: This category of methods incorporates user information as the context within the prompts for LLMs. By providing this contextual information, LLMs can either directly perform downstream personalization tasks through text generation or act as intermediate modules to extract more relevant information, thereby enhancing the system's performance on downstream tasks.

- **Personalization via Representation Learning (Sec 5.3)**: This category of methods encodes user information into the embedding spaces of neural network modules. The user information can be represented through the entire parameters of the LLM, a subset of the model's parameters, a small number of additional parameters, or an explicit embedding vector specific to each user.

- **Personalization via RLHF (Sec 5.4)**: This category of methods uses user information as the reward signal to align LLMs with personalized preferences through reinforcement learning.

The following sections describe different techniques for achieving personalization in downstream tasks. Note that most of these approaches are orthogonal to each other, meaning they can coexist within the same system. We provide a summary of the personalization techniques organized intuitively using the proposed taxonomy in Table 3.

### 5.1 Personalization via Retrieval-Augmented Generation

Retrieval-augmented generation (RAG) enhances LLM performance by retrieving relevant document segments from an external knowledge base using semantic similarity calculations (Gao et al., 2023). This approach is widely employed in information retrieval and recommendation systems (Zhao et al., 2024b; Rajput et al., 2023; Di Palma, 2023; Wang et al., 2024g). While RAG can reduce hallucinations by grounding the generation process in retrieved factual content (Shuster et al., 2021; Li et al., 2024d), these retrieval modules can also be used to retrieve personalized information, enabling the generation of customized, tailored outputs.

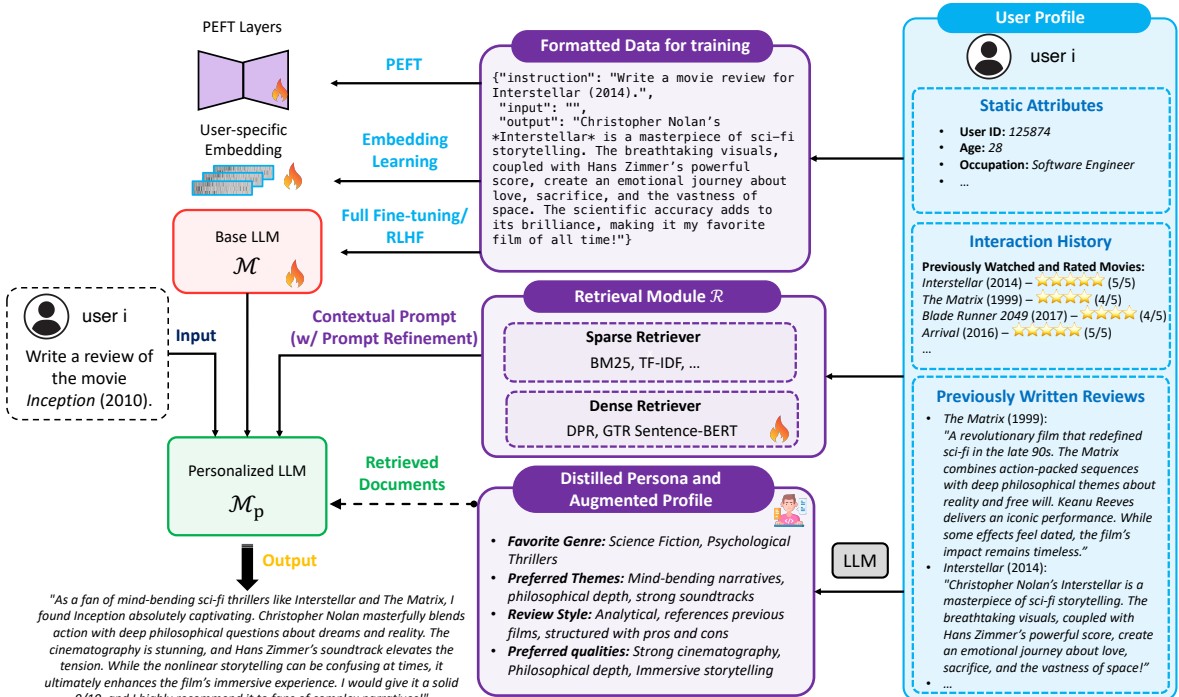

Figure 7: A Case Study on Personalized Movie Review Generation. This figure illustrates how different personalization techniques for LLMs enhance personalized movie review generation by leveraging user profiles, retrieval modules, and fine-tuning methods to align outputs with individual preferences and writing styles.

**Definition 12** (RETRIEVAL MODEL). *A Retrieval Model $\mathcal{R}$ is a system designed to identify and return relevant information from a large external database $\mathcal{D}$ in response to a query $q \in \mathcal{Q}$. Given a query $q$, the retrieval model aims to find the document or data point $d^* \in \mathcal{D}$ that maximizes the relevance function $r(q, d)$:*

$$d^* = \arg\max_{d \in \mathcal{D}} r(q, d)$$

*where $d$ represents an individual document or data point in $\mathcal{D}$.*

**Definition 13** (RETRIEVAL-AUGMENTED GENERATION). Retrieval-augmented Generation *(RAG) is a process in which a language model $\mathcal{M}$ leverages a retrieval model $\mathcal{R}$ to enhance its generation capabilities. Given an input $X_u$ from user $u$, the retrieval model $\mathcal{R}$ identifies $k$ relevant external data points or documents from a dataset $\mathcal{D}$. These retrieved data points are then incorporated into the input text to form a transformed input $\bar{x}$, which is used by the language model $\mathcal{M}$ to generate a grounded output $\hat{y}_u$.*

*Formally, the process can be described as follows:*

$$\bar{x} = \phi_p(X_u, \mathcal{R}(\phi_q(X_u), \mathcal{D}, k))$$

$$\hat{y}_i = M(\bar{x})$$

*where $\phi_q$ is a query construction function used by the retrieval model $\mathcal{R}$ to find relevant documents, and $\phi_p$ is a prompt construction function that integrates the retrieved information into the original input $X_u$. The output $\hat{y}_u$ represents the generated text based on both the original input and the retrieved information.*

For personalization tasks, large user profiles often serve as external knowledge bases since they cannot be fully incorporated into prompts due to LLMs' context limitations. As a result, RAG is commonly employed in personalized LLM systems. In this section, we discuss and categorize the personalization techniques that utilize RAG. We categorize those RAG-based personalization techniques based on the retriever into the following two main categories:

- **Sparse Retrieval (Sec. 5.1.1)**: This category of methods employs frequency-based vectors to encode queries and user information, which are then used for retrieval in downstream personalization tasks. Since this approach only requires statistical computations such as frequency counts, it is highly efficient. These methods demonstrate robust performance in information retrieval tasks, frequently serving as baselines in RAG systems.

- **Dense Retrieval (Sec. 5.1.2)**: This category of methods employs deep neural networks including LLM-based encoders to generate continuous embeddings for queries and documents in retrieval tasks. These encoding layers can either use off-the-shelf models directly for downstream tasks without tuning their parameters or incorporate trainable parameters that can be adjusted specifically for retrieval tasks.

Another retrieval method, such as black-box retrieval, involves using external APIs like Google or Bing, which are commonly integrated into LLM-based agent frameworks via tool-use. While this can be valuable for personalization in specific scenarios, we do not explore it in detail due to its black-box nature, which limits transparency regarding how user information is utilized and how personalization is achieved. Additionally, this design tends to be highly tool-specific, reducing its generalizability. It is also worth noting that many approaches also employ hybrid methods that combine elements of both sparse and dense retrieval.

### 5.1.1 Sparse Retrieval

Sparse retrieval encodes both queries and documents as sparse vectors, typically based on word frequency and importance. It operates by matching terms in the query with terms in the document, focusing on exact term overlap. Due to its simplicity and effectiveness, sparse retrieval has long been a foundational approach in information retrieval systems. The two most commonly used examples of sparse retrievers are TF-IDF (term frequency-inverse document frequency) (Sparck Jones, 1972) and BM25 (Best Matching 25) (Robertson et al., 1995).

**TF-IDF:** This method scores documents based on the frequency of terms relative to their occurrence across the entire document collection. It is calculated as:

$$\text{TF-IDF}(q_i, D) = \text{TF}(q_i, D) \cdot \text{IDF}(q_i)$$

where the term frequency (TF) of term $q_i$ in document $D$ is:

$$\text{TF}(q_i, D) = \frac{f(q_i, D)}{|D|}$$

and the inverse document frequency (IDF) is:

$$\text{IDF}(q_i) = \log \frac{N}{n(q_i)}$$

Here, $f(q_i, D)$ is the frequency of term $q_i$ in document $D$, $|D|$ is the total number of terms in $D$, $N$ is the total number of documents in the collection, and $n(q_i)$ is the number of documents containing the term $q_i$. To prevent division by zero and dampen the effect of very rare terms, different smoothed versions are often used.

**BM25:** This is a more advanced sparse retrieval method that extends the TF-IDF model by incorporating document length normalization and saturation controls for term frequency. The BM25 score for a document $D$ with respect to a query $q_i$ is calculated as:

$$\text{BM25}(q_i, D) = \text{IDF}(q_i) \cdot \frac{f(q_i, D) \cdot (k_1 + 1)}{f(q_i, D) + k_1 \cdot (1 - b + b \cdot \frac{|D|}{\text{avgdl}})}$$

where $q_i$ is the $i$-th query term, $f(q_i, D)$ is the term frequency of $q_i$ in document $D$, $|D|$ is the length of document $D$, avgdl is the average document length in the collection, and $k_1$ and $b$ are parameters that control

term frequency saturation and document length normalization, respectively. Typical values are $k_1 = 1.2$ and $b = 0.75$.

Because of their generalizability, effectiveness, and simplicity, sparse retrievers often serve as baselines in retrieval-based personalization methods (Salemi et al., 2023; Li et al., 2023b; Richardson et al., 2023). For instance, Salemi et al. (2023) use BM25 as one of the retrievers to fetch relevant user information, which is then incorporated into prompts for LLMs when evaluating on the LaMP dataset (Salemi et al., 2023). Richardson et al. (2023) enhance personalization in LLMs by integrating BM25 retrieval with user data summaries, achieving improved performance on LaMP tasks while reducing the volume of retrieved data by 75%. In another work, Li et al. (2023b) propose a multistage, multitask framework inspired by writing education to enhance personalized text generation in LLMs. In the initial retrieval stage, BM25 is used to retrieve relevant past user documents, which are subsequently ranked and summarized to generate personalized text.

Sparse retrieval serves as a foundation for many personalized systems, particularly in scenarios where large amounts of user information are involved, and efficiency is crucial. However, while sparse retrieval techniques like BM25 and TF-IDF excel in general retrieval tasks, they have inherent limitations in personalization. The lexical matching nature of these methods struggles to capture semantic relationships between terms, which can hinder their performance in complex personalization tasks. This issue is especially relevant in scenarios where user preferences or behaviors require deeper understanding beyond keyword overlap.

### 5.1.2 Dense Retrieval

Dense retrievers leverage deep neural networks to generate continuous representations for queries and documents, enabling retrieval in a dense embedding space via similarity-based search (Johnson et al., 2021). Some works (Sun et al., 2024) employ pre-trained LLM encoders, such as OpenAI's `text-embedding-ada` series and Sentence-BERT (Reimers & Gurevych, 2019), without fine-tuning their parameters. Other approaches focus on training retrieval-oriented embeddings. For example, Dense Passage Retrieval (DPR) (Karpukhin et al., 2020) employs dense embeddings in a dual-encoder framework, using BM25 hard negatives and in-batch negatives to efficiently retrieve relevant passages for open-domain question answering. Contriever (Izacard et al., 2021) is an unsupervised dense retriever trained using contrastive learning, where two random spans from a document are cropped independently to form positive pairs for training. In the context of personalization, several works have proposed specialized training data construction (Mysore et al., 2023a;b) and training strategies (Salemi et al., 2023; 2024) to enhance dense retrievers' ability to retrieve more relevant user information, improving personalization for downstream tasks using LLMs. Salemi et al. (2023) use Fusion-in-Decoder (Izacard & Grave, 2021) on encoder-decoder models such as T5 (Raffel et al., 2020) which retrieves and concatenate multiple relevant documents' encoded embedding before decoding. Mysore et al. (2023a) train a pre-trained MPNET model (Song et al., 2020) with a scale-calibrating KL-divergence objective which shows superior performance on personalized open-ended long-form text generation tasks. Salemi et al. (2024) train the Contriever model for personalizing LLMs using LLMs' feedback with policy gradient optimization and knowledge distillation. Zeng et al. (2023) introduce UIA, a flexible dense retrieval framework that incorporates personalized attentive networks to enhance various information access such as keyword search, query by example, and complementary item recommendation. Other dense retrievers such as Sentence-T5 (Ni et al., 2021) and Generalizable T5-based dense Retrievers (GTR) (Ni et al., 2022) are also frequently used for downstream personalization tasks. Generally, while dense retrievers require training on downstream tasks, making them more costly and time-inefficient (Richardson et al., 2023), they tend to achieve superior performance compared to sparse retrievers in downstream personalization tasks. However, constructing effective training data, designing suitable loss functions, and incorporating LLMs into the training process to optimize retrievers for improved downstream personalization tasks remain open challenges.

### 5.2 Personalization via Prompting

**Definition 14** (PROMPT ENGINEERING)**.** Prompt Engineering *is the process of designing, refining, and optimizing prompts to achieve desired outputs from language models. This involves iterative testing and adjustment of prompts to enhance the model's performance on various tasks, improve response accuracy, and align the model's outputs with user expectations or specific application requirements.*

A prompt serves as an input for a Generative AI model, guiding the content it generates (Meskó, 2023; White et al., 2023; Heston & Khun, 2023; Hadi et al., 2023; Brown et al., 2020; Schulhoff et al., 2024). Empirically, better prompts enhance LLMs' performances across a wide range of tasks (Wei et al., 2022b; Liu et al., 2023c). As a result, there has been a substantial increase in research dedicated to designing more effective prompts to achieve better outcomes, a field known as prompt engineering. In this section, we categorize personalization techniques that leverage prompt engineering into three main categories:

- **Contextual Prompting (Sec. 5.2.1)**: These methods directly incorporate user history information into the prompt, enabling LLMs to perform downstream personalization tasks based on this contextual data.

- **Persona-based Prompting (Sec. 5.2.2)**: These approaches introduce specific personas, such as demographic information, into the prompt. By encouraging LLMs to role-play these personas, it aims to enhance the performance of downstream personalization tasks.

- **Profile-Augmented Prompting (Sec. 5.2.3)**: These methods focus on designing prompting strategies that enrich the original user history information by leveraging LLMs' internal knowledge, thereby improving downstream personalization tasks.

- **Prompt Refinement (Sec. 5.2.4)**: This category of methods focuses on developing robust frameworks that iteratively refine the initial hand-crafted prompts, enhancing downstream personalization.

### 5.2.1 Contextual Prompting

As current LLMs demonstrate increasing abilities and extended context lengths (Jin et al., 2024a; Ding et al., 2024; Lin et al., 2024b), one naive approach is to directly include a proportion of past user information in the prompt and ask the LLMs to predict user behavior on downstream tasks (Di Palma et al., 2023; Wang & Lim, 2023; Sanner et al., 2023; Li et al., 2023e; Christakopoulou et al., 2023). For example, Kang et al. (2023) investigate the performance of multiple LLMs in user rating prediction tasks by directly incorporating the user's past rating history and candidate item features in a zero-shot and few-shot manner. This work finds that LLMs underperform traditional recommender systems in zero-shot settings but achieve comparable or superior results when fine-tuned with minimal user interaction data. Larger models (100B+ parameters) show better performance and faster convergence, highlighting LLMs' data efficiency and potential in recommendation tasks. Similarly, Liu et al. (2023a) investigate the potential of ChatGPT as a general-purpose recommendation model by directly injecting user information in the prompt and evaluating its performance on five recommendation tasks: rating prediction, sequential recommendation, direct recommendation, explanation generation, and review summarization. The study finds that while ChatGPT performs well in generating explanations and summaries, it shows mixed results in rating prediction and poor performance in sequential and direct recommendations, indicating the need for further exploration and improvements. These studies suggest that directly incorporating past user information into LLM prompts as contextual input could be a promising solution for a wide range of personalized downstream tasks. However, this approach faces challenges in scalability when dealing with large, unstructured user data, as LLMs may struggle to interpret such data effectively (Liu et al., 2024b). Additionally, while it offers improved explainability, it may not achieve significant performance gains over traditional non-LLM-based methods.

### 5.2.2 Persona-based Prompting

LLMs have been widely used for role-playing and imitating human behavior, mainly by specifying the desired persona within the prompt (Aher et al., 2023; Horton, 2023; Kovač et al., 2023; Argyle et al., 2023; Dillion et al., 2023; Woźniak et al., 2024; Li et al., 2024e). Generally, the persona denotes the entity whose viewpoints and behaviors the simulation seeks to examine and reproduce. This persona can encompass relatively stable characteristics (e.g., race/ethnicity), those that gradually evolve (e.g., age), or those that are transient and situational (e.g., emotional state) (Yang, 2019). Chen et al. (2024b) categorize persona into three types: *demographic persona*, which represents aggregated characteristics of demographic segments (Huang et al., 2023a; Xu et al., 2023; Gupta et al., 2023); *character persona*, encompassing well-established characters from real and fictional sources (Shao et al., 2023; Wang et al., 2023d; 2024f); and *individualized persona*, constructed from individual behavioral and preference data to provide personalized services which is discussed in Sec.

5.2.1. For example, to induce an extroverted persona in LLMs, Jiang et al. (2023) use the prompt of "*You are a very friendly and outgoing person who loves to be around others. You are always up for a good time and love to be the life of the part*" which achieve more consistent results on human psychological assessments such as the Big Five Personality Test (Barrick & Mount, 1991). Through this type of prompt construction, LLMs can deviate from their intrinsic "personality" (Karra et al., 2022; Safdari et al., 2023; Huang et al., 2023b; Santurkar et al., 2023; Hartmann et al., 2023) and exhibit altered characteristics in their responses in accordance with the requirements in the prompt. Another approach involves using LLMs to mimic well-known figures (e.g., Elon Musk). Prior works achieve this by incorporating descriptions of character attributes—such as identity, relationships, and personality traits—or by providing demonstrations of representative behaviors that reflect the characters' linguistic, cognitive, and behavioral patterns within the prompt (Han et al., 2022; Li et al., 2023a; Chen et al., 2023a; Zhou et al., 2023; Shen et al., 2023; Yuan et al., 2024; Chen et al., 2024a). While persona-based role-playing can effectively reflect certain personalities, potentially improving adaptive personalization by dynamically adjusting to user-specific behaviors and preferences over time, it also raises significant concerns. 'Persona prompting' may lead to issues such as 'character hallucination,' where the model exhibits knowledge or behavior misaligned with the simulated persona. Additionally, it may introduce biases (Gupta et al., 2023; Zhang et al., 2023d; Wang et al., 2024a; Ziems et al., 2024), toxicity (Deshpande et al., 2023), potential jailbreaking (Chao et al., 2023; Liu et al., 2023g; Chang et al., 2024; Xu et al., 2024b), ecological fallacy (Orlikowski et al., 2023), and susceptibility to caricatures (Cheng et al., 2023b), among other risks.

### 5.2.3 Profile-Augmented Prompting

In many personalization datasets, there are two main issues with the user-profile database. First, the size of the user data is often so large that it may exceed the model's context length or contain a significant amount of irrelevant information, which can distract the model (Shi et al., 2023; Liu et al., 2024c). Second, despite the large size, the user-profile database frequently contains incomplete or insufficient information (Perez et al., 2007; Dumitru et al., 2011) and sparse user interactions (e.g., cold-start). For example, movie recommendation datasets typically only contain the main actors and a brief plot summary, but this often overlooks key details like genre, tone, and thematic depth, leading to less effective recommendations. Another example could be 'lurkers,' users with minimal interaction history, a common scenario in recommendation systems that makes it difficult to give personalized responses. To resolve these problems, a line of work focuses on eliciting the internal knowledge of LLMs to augment or distill existing user profiles (Zheng et al., 2023b; Wu et al., 2024b). Richardson et al. (2023) propose a summary-augmented approach by extending retrieval-augmented personalization with task-aware user summaries generated by LLMs in the prompt. ONCE (Liu et al., 2024d) generates user profiles by prompting LLMs to summarize the topics and regions of interest extracted from their browsing history, which helps LLMs capture user preferences for downstream tasks. Lyu et al. (2023) propose LLM-REC, which employs four prompting strategies to augment the original item descriptions, which often contain incomplete information for recommendation. These augmented descriptions are then concatenated as the input for the following recommendation module, introducing the relevant context and helping better align with user preferences. Xi et al. (2023) use factorization prompting to extract nuanced user preferences and item details from user profiles, and employs a hybrid-expert adaptor to convert this knowledge into augmented vectors for existing recommendation models. Sun et al. (2024) introduce Persona-DB, a hierarchical construction of user personas from interaction histories through prompting LLMs, and a collaborative refinement process that integrates data from similar users to enhance the accuracy and efficiency of personalized response forecasting.

### 5.2.4 Prompt Refinement

Most works using prompt engineering in personalization tasks rely on hand-crafted prompts, which necessitate human expertise and can be costly, with their effectiveness being verifiable only through trial and error. Some research efforts aim to train models to refine these manually designed prompts, enhancing their capability for personalization. In the context of personalization, Li et al. (2024a) train a small LM such as T5 (Raffel et al., 2020) for revising text prompts and enhancing personalized text generation with black-box LLMs. Their approach combines supervised learning and reinforcement learning to optimize prompt rewriting. Kim & Yang (2024) propose FERMI (Few-shot Personalization of LLMs with Mis-aligned Responses), a method that

optimizes input prompts by iteratively refining them based on user profiles and past feedback, while also incorporating contexts of misaligned responses to enhance personalization. The optimization process involves three steps: scoring the prompts based on user feedback, updating a memory bank of high-scoring prompts and their contexts, and generating new, improved prompts. Additionally, the method refines inference by selecting the most relevant personalized prompt for each test query, leading to significant performance improvements across various benchmarks (Santurkar et al., 2023; Durmus et al., 2023; Salemi et al., 2023).

## 5.3 Personalization via Representation Learning

Personalized representation learning aims to learn latent representations that accurately capture each user's behavior, with applications in personalized response generation, recommendations, etc (Li & Zhao, 2021; He et al., 2023; Tan & Jiang, 2023). In this section, we discuss and categorize personalization techniques that leverage representation learning into the following main categories:

- **Full-Parameter Fine-tuning (Sec. 5.3.1)**: This category of methods focuses on developing training strategies and curating datasets to update all parameters of the LLM, enhancing its ability to perform downstream personalization tasks more effectively.

- **Parameter-Efficient Fine-tuning (PEFT) (Sec. 5.3.2)**: This category of methods avoids fine-tuning all the parameters by updating only a small number of additional parameters or a subset of the pretrained parameters to adapt the LLMs to downstream personalization tasks. This selective tuning allows for the efficient encoding of user-specific information.

- **Embedding Learning (Sec. 5.3.3)**: This category of methods focuses on learning embeddings that represent both input text and user information in vectorized form, enabling models to more effectively incorporate personalized features and preferences into the learning process.

### 5.3.1 Full-Parameter Fine-tuning

**Definition 15** (FINE-TUNING)**.** Fine-tuning *is the process of adapting a LLM $\mathcal{M}$ to a specific task by further training it on a smaller, targeted dataset $\mathcal{D}_i$ after the pre-training stage. This updates the model's parameters $\theta$ to improve performance on the specified downstream tasks. Formally, fine-tuning can be expressed as:*

$$\theta^* = \arg\min_{\theta} \mathcal{L}(\mathcal{M}(X_i; \theta), Y_i)$$

*where $X_i$ is the input data, and $Y_i$ is the corresponding output. The fine-tuned model $\mathcal{M}$ with parameters $\theta^*$ is optimized for generating desired responses.*

The "pre-train, then fine-tune" paradigm has been widely adopted, enabling the development of foundation models that can be adapted to a range of applications after acquiring general knowledge from large corpus pre-training (Bommasani et al., 2021; Min et al., 2023; Liu et al., 2023c). Fine-tuning LLMs for targeted scenarios generally yields better results on most tasks when model parameters are accessible and the associated costs are acceptable (Gao et al., 2023). For instance, fine-tuning allows LLMs to adapt to specific data formats and generate responses in a particular style as instructed, which is crucial for many personalization tasks (Du & Ji, 2022). Empirically, it can achieve better performances on some personalization tasks than zero-shot or few-shot prompting off-the-shell LLMs (Kang et al., 2023). Li et al. (2023b) use an auxiliary task, called author distinction, to train a T5-11B model (Raffel et al., 2020) to better distinguish if two documents are from the same user to obtain a better-personalized representation. Yin et al. (2023) first use ChaGPT to fuse heterogeneous user information through prompt engineering to build an instructional tuning dataset and use this dataset to fine-tune ChatGLM-6B (Du et al., 2021), resulting in enhanced recommendation performance. In this line of work, a common approach involves providing a task instruction that includes user interaction history and a potential item candidate, with a label of 'Yes' or 'No' indicating whether the user prefers the item. Fine-tuning LLMs, such as LLaMA (Touvron et al., 2023), on such instruction-tuning datasets generally yields superior performance on recommendation tasks compared to both prompting LLMs and traditional methods (Hidasi et al., 2015). Yang et al. (2023) fine-tune a LLaMa 7B with an instruction tuning dataset that provides instructions for generating a list of "future" items the user may interact with, based on a list of past interactions, or for retrieving target "future" items from a list of candidate items.

### 5.3.2 Parameter-Efficient Fine-tuning

**Definition 16** (PARAMETER-EFFICIENT FINE-TUNING)**.** *Parameter-efficient Fine-tuning (PEFT) is a technique for adapting LLMs to specific tasks by updating only a small subset of the model's parameters $\theta_t \subset \theta$ or introducing a limited set of new parameters $\theta_{new}$, while keeping the majority of the original parameters $\theta$ unchanged.*

*Formally, given an LLM $\mathcal{M}$ with parameters $\theta$, PEFT seeks to minimize the loss function $\mathcal{L}$ over a reduced set of parameters:*

$$\theta_t^*, \theta_{new}^* = \arg \min_{\theta_t, \theta_{new}} \mathcal{L}\left(\mathcal{M}(X_i; \theta_t, \theta_{new}, \theta_{frozen}), Y_i\right)$$

*where $X_i$ is the input data, $Y_i$ is the corresponding output, and $\theta_{frozen}$ represents the parameters that remain fixed during fine-tuning, and $\theta_{new}$ may be empty if no new parameters are introduced.*

Tan et al. (2024b) introduce One PEFT Per User (OPPU), a method that employs personalized PEFT modules such as LoRA (Hu et al., 2021) and prompt tunning parameters (Lester et al., 2021) to encapsulate user-specific behavior patterns and preferences. Based on OPPU, Tan et al. (2024a) further propose PER-PCS, a framework enabling efficient and fine-grained personalization of LLMs by allowing users to share and assemble personalized PEFT pieces collaboratively. Dan et al. (2024) introduce P-Tailor which personalizes LLMs by using a mixture of specialized LoRA experts to model the Big Five Personality Traits. Huang et al. (2024b) propose Selective Prompt Tuning which improves personalized dialogue generation by adaptively selecting suitable soft prompts for LLMs based on input context.

### 5.3.3 Embedding Learning

**Definition 17** (EMBEDDING)**.** *An* Embedding *is a vector in a continuous space produced by an embedding function $Emb(\cdot)$, which maps discrete data, such as tokens, into continuous vector spaces. This transformation allows text to be represented in a format suitable for machine learning models. Given a token $w$, the embedding function produces a vector $\mathbf{e} \in \mathbb{R}^d$, where $d$ is the dimensionality of the embedding space:*

$$\mathbf{e} = Emb(w)$$

**Definition 18** (USER-SPECIFIC EMBEDDING LEARNING)**.** *User-specific Embedding Learning involves creating embeddings that capture individual user preferences and behaviors from their interaction data. These embeddings are used to personalize model outputs.*

*Formally, given user interactions $I_i$, the embedding $\mathbf{e}_i$ is obtained via:*

$$\mathbf{e}_i = Emb(I_i)$$

*The embedding is then integrated into the model to generate personalized responses:*

$$g_i = \mathcal{M}(X_i; \mathbf{e}_i)$$

*where $X_i$ is the input data, and $\mathcal{M}$ is the model. This approach enhances personalization by adapting the model's responses to individual user characteristics.*

Cho et al. (2022) present a personalized dialogue generator that detects an implicit user persona using conditional variational inference to produce user-specific responses based on dialogue history, enhancing user engagement and response relevance. HYDRA (Zhuang et al., 2024) enhances black-box LLM personalization by capturing user-specific behavior patterns and shared general knowledge through model factorization. It employs a two-stage retrieve-then-rerank workflow and trains an adapter to align model outputs with user-specific preferences. Ning et al. (2024b) propose a USER-LLM that leverages user embeddings to dynamically contextualize LLMs. These user embeddings are distilled from diverse user interactions using self-supervised pretraining, capturing latent user preferences and their evolution over time. By integrating these embeddings through cross-attention and soft-prompting, the approach enhances personalization and performance across various tasks while maintaining computational efficiency. Liu et al. (2024a) propose the PPlug model, which personalizes LLMs by employing a lightweight plug-in user embedder that aggregates user

historical behaviors into a single, input-aware embedding. This embedding guides a fixed LLM to generate personalized outputs without modifying the model's parameters, significantly improving personalization performance over retrieval-based methods.

### 5.4 Personalization via Reinforcement Learning from Human Feedback (RLHF)

LLMs are learned in multiple stages (Ouyang et al., 2022): pre-training on large amounts of text, fine-tuning on domain-specific data, and alignment to human preferences. While the alignment is usually done to capture general user preferences and needs, it can also be used for personalization, to align to expectations and requirements of individual users. The set of algorithmic techniques used for alignment is know as *Reinforcement Learning from Human Feedback (RLHF)* and we review these techniques next.

In classic *reinforcement learning (RL)* (Sutton & Barto, 1998), an agent learns a policy to optimize a long-term goal from reward signals. This can be done by directly learning a policy (Williams, 1992; Baxter & Bartlett, 2001; Schulman et al., 2015; 2017), learning a value function (Bellman, 1957; Sutton, 1988), or a combination of both (Sutton et al., 2000). In RLHF, the agent learns a policy represented by an LLM (Ouyang et al., 2022; Ahmadian et al., 2024; Rafailov et al., 2024; Xu et al., 2024a), which is optimized based on preferential human feedback on its outputs. The preferential feedback is rooted in social sciences, and can be binary (Bradley & Terry, 1952) or over multiple options (Plackett, 1975; Zhu et al., 2023a). RLHF helps the LLM to align with human values, promoting more ethically sound and socially responsible AI systems (Kaufmann et al., 2023). The first methods for aligning LLMs through RLHF were based on learning a proxy reward model from human feedback (Nakano et al., 2021; Ouyang et al., 2022; Bai et al., 2022; Dubois et al., 2024; Lin et al., 2024a; Chakraborty et al., 2024). This can be viewed as a form of reward shaping (Ng et al., 1999) applied to a reward model that captures general preferences of a population. Modern alignment techniques (Rafailov et al., 2024) optimize the LLM directly from human feedback.

Besides general preferences, some works (Jang et al., 2023) also investigate the alignment of LLMs with personalized human preferences. The motivation for this task is that even for the same prompt, different users may desire different outputs, and individual preferences can vary across different dimensions (Casper et al., 2023). For example, when asked "What is an LLM?", a PhD student in NLP may prefer a detailed technical explanation, while a non-expert might seek a simplified and concise definition. Jang et al. (2023) frame this problem as a Multi-Objective Reinforcement Learning (MORL) task, where diverse and potentially conflicting user preferences are decomposed into multiple dimensions and optimized independently. It can be efficiently trained independently and combined effectively post-hoc through parameter merging. Chen et al. (2024d) introduce Personalized Alignment at Decoding-Time (PAD), a training-free framework for aligning large language models with personalized user preferences during inference. PAD employs a personalized reward modeling strategy that decouples text generation dynamics from user preferences, enabling generalizable token-level personalized rewards to guide the decoding process. This approach dynamically adjusts model outputs to diverse or unseen preferences without requiring retraining, demonstrating scalability and superior alignment performance across multiple base models. Li et al. (2024g) propose a Personalized-RLHF (P-RLHF) framework, where user-specific models are jointly learned alongside language or reward models to generate personalized responses based on individual user preferences. The method includes developing personalized reward modeling (P-RM) and personalized Direct Preference Optimization (P-DPO) objectives, tested on text summarization data, demonstrating improved alignment with individual user-specific preferences compared to non-personalized models. Park et al. (2024a) address the challenge of heterogeneous human preferences in RLHF, using personalized reward models through representation learning and clustering, as well as preference aggregation techniques grounded in social choice theory and probabilistic opinion pooling. Kirk et al. (2024) introduce the PRISM Alignment Project which is a novel dataset that maps the sociodemographics and preferences of 1,500 diverse participants from 75 countries onto their feedback in over 8,000 live conversations with 21 LLMs. This dataset aims to enhance the alignment of AI systems by incorporating wide-ranging human perspectives, especially on value-laden and controversial topics. Lee et al. (2024) propose an approach to aligning LLMs with diverse human preferences through system message generalization, leveraging a comprehensive dataset named Multifaceted Collection to train the model JANUS, which effectively adapts to a wide range of personalized user preferences. Yang et al. (2024a) propose Rewards-in-Context which fine-tunes LLMs by conditioning responses on multiple rewards during supervised fine-tuning, enabling flexible

preference adaptation at inference time. This method achieves Pareto-optimal alignment across diverse objectives, using significantly fewer computational resources compared to traditional MORL approaches. Poddar et al. (2024) propose Variational Preference Learning (VPL), which is a technique to personalize RLHF by aligning AI systems with diverse user preferences through a user-specific latent variable model. VPL incorporates a variational encoder to infer a latent distribution over hidden user preferences, enabling the model to condition its reward functions and adapt policies based on user-specific context. In simulated control tasks, VPL demonstrates effective modeling and adaptation to diverse preferences, with enhanced performance and personalization capabilities compared to standard RLHF methods. Another idea is to offer demonstrations as an efficient alternative to pairwise preferences, allowing users to directly showcase their desired behavior through example completions or edits. This method, as demonstrated by DITTO (Shaikh et al., 2024), enables rapid and fine-grained alignment to individual preferences with minimal data, making it a potentially powerful tool for personalizing LLMs with limited user input.

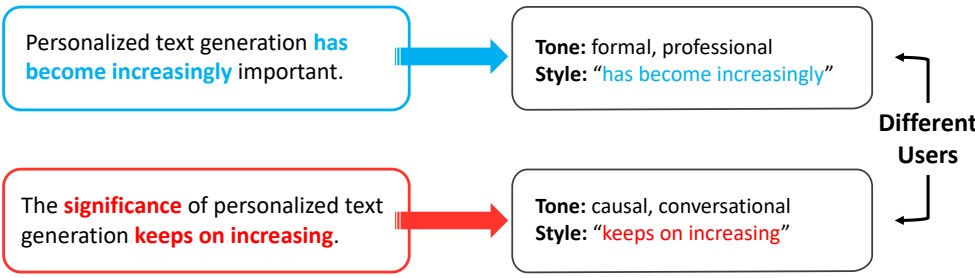

Figure 8: **Personalization of User-specific Writing Style and Tone.**

## 5.5 Discussion

Here, we discuss the computational and scalability considerations, effectiveness comparisons, and practical guidance for implementing various personalization techniques for LLMs. From a computational and scalability perspective, personalization strategies vary significantly in their resource demands and implementation complexity. RAG-based methods, especially when relying on dense retrievers, can deliver strong personalization fidelity (Salemi et al., 2023) but may incur considerable indexing overhead and GPU memory usage (Asai et al., 2023; Fan et al., 2024). This has led to growing interest in hybrid approaches that integrate sparse methods (e.g., BM25) for greater efficiency with dense retrieval techniques to maintain semantic alignment (Novotný & Stefánik, 2022; Arabzadeh et al., 2021). Prompting-based methods generally do not require large-scale databases or model parameter tuning, making them computationally efficient and lightweight. However, deriving effective personas from extensive user data or implementing multi-level hierarchical prompting (Sun et al., 2024) may involve iterative model inference, introducing latency challenges in real-world applications. In the future, distillation-based approaches (Li et al., 2023c), which transfer the general capabilities of large LLMs into smaller, task-specific models tailored for personalization, could potentially enhance efficiency by preserving personalization fidelity while reducing computational costs. Other techniques such as caching pipelines (Wu et al., 2024c) and on-demand personalization for high-value users also shows promising as practical solutions to reduce inference latency and memory consumption. Additionally, federated or on-device personalization offers a promising avenue to alleviate server-side resource burdens while ensuring user privacy (Kulkarni et al., 2020; Li et al., 2021). RLHF-based methods, while capable of delivering high-quality personalization, often require extensive and costly human feedback collection, which can be prone to noise (Casper et al., 2023; Kaufmann et al., 2023). Alternatives such as Reinforcement Learning from AI Feedback (RLAIF) (Lee et al.) may reduce costs but introduce challenges, such as limitations in capturing nuanced user preferences and ensuring feedback quality. In terms of effectiveness, recent benchmarks like LamP (Salemi et al., 2023) and longLamP (Kumar et al., 2024) indicate that methods leveraging user-specific data retrieval (e.g., BM25, Contriever) generally excel in tasks that demand explicit grounding, with dense retrievers proving especially adept at complex abstractions. Prompt-based personalization remains highly flexible and lightweight in scenarios with moderate amounts of user data, yet its performance degrades as context windows become saturated. Meanwhile, Kumar et al. (2024) find that representation learning approaches such as

fine-tuning FLAN-T5 or incorporating user embeddings tend to yield deeper personalization gains, though they risk overfitting when per-user data is sparse and require substantial computational resources for training. As a result, deciding which technique to employ depends largely on application requirements and constraints. RAG is well-suited to large knowledge bases or dynamic user profiles where context must be retrieved on the fly, while prompting methods are preferable for conversational systems that can accommodate brief historical interactions without retraining. Organizations with abundant, high-quality user data and significant computational capacity may consider representation learning, particularly when nuanced personalization is important. RLHF demonstrates significant promise in scenarios where continuous alignment with user feedback is essential, particularly when pairwise preference data is readily available. However, the choice of personalization method is shaped by multiple factors, including the availability of data, infrastructure constraints, latency requirements, and regulatory considerations. These diverse influences ensure that no single personalization approach is universally optimal across all tasks, highlighting the need for context-specific strategies.

## 6    Taxonomy of Evaluation Metrics for Personalized LLMs

In this section, we introduce a taxonomy for the evaluation of personalized LLM techniques. In particular, we categorize evaluation as *intrinsic evaluation* of the personalized text generated (Sec. 6.1) or *extrinsic evaluation* that relies on a downstream application such as recommendation systems to demonstrate the utility of the generated text from the personalized LLM (Sec. 6.2).

The performance of a *personalized LLM* for a downstream application (such as the generation of an email personalized for a specific user or the generation of an abstract by a specific user) should be quantified using an appropriate evaluation metric. As an example, for the direct evaluation task of personalized text generalization, there are many facets that must be considered for the personalized evaluation of the generated text for a specific user. There are largely the following factors that are illustrated in Figure 4 and Table 1. Figure 8 illustrates an example on how the same information can be personalized and expressed in different writing styles and tones, which should be measured by the employed metrics.

**Definition 19** (EVALUATION METRIC). *For an arbitrary dataset $\mathcal{D}$, there is a subset of evaluation metrics $\psi(\mathcal{D}) \subseteq \Psi$ that can be used for $\mathcal{D}$, where $\Psi$ is the space of all metrics and $\psi(\mathcal{D})$ is the subset of metrics appropriate for the dataset $\mathcal{D}$.*

**Definition 20** (INTRINSIC EVALUATION). *Intrinsic Evaluation $\mathbb{E}_i$ refers to the assessment of personalized text generated by an LLM $\mathcal{M}_p$ based on predefined metrics $\psi(\cdot) \in \Psi$ that measure the quality, relevance, and accuracy of the generated content $\hat{y} \in \hat{\mathbb{Y}}$ against ground-truth data $Y \in \mathbb{Y}$. This evaluation is performed directly on the output of the model:*

$$\mathbb{E}_i(\hat{y}, Y) = \psi(\hat{y}, Y)$$

*Note that the ground-truth data $Y$ can represent user-written text when available or, alternatively, user preferences and reward models that capture user judgments.*

**Definition 21** (EXTRINSIC EVALUATION). *Indirect Evaluation $\mathbb{E}_e$ involves assessing the utility of the personalized text generated by an LLM $\mathcal{M}_p$ through its impact on a downstream application $\mathcal{F}$. The evaluation measures the effectiveness of the generated content by comparing the predictions $\hat{r}$ with the ground-truth labels $r$ using application-specific metrics:*

$$\mathbb{E}_e(\hat{r}, r) = \psi_a(\hat{r}, r)$$

*where $\psi_a(\cdot) \in \Psi_a$ represents the application-specific metrics.*

More formally, let $\mathbb{E}_i$ represent intrinsic evaluation and $\mathbb{E}_e$ represent extrinsic evaluation. Let $\hat{y} = \mathcal{M}(X; \theta)$ denote the generated content for input $X$ from dataset $D$, and let $Y$ represent the ground truth output from $D$. Similarly, let $\hat{r} = \mathcal{F}(\hat{y})$ be the downstream task predictions based on $\hat{y}$, and $r$ be the ground truth output for the downstream tasks.

- **Intrinsic Evaluation (Sec. 6.1)**: Formally, intrinsic evaluation metrics can be defined as $\psi(\mathcal{D}) = \{\mathbb{E}_i \mid \mathbb{E}_i(\hat{y}, Y)\}$. Most research on personalized text generation focuses on scenarios where ground-truth user-written text is available (Salemi et al., 2023; Kumar et al., 2024). In this setting, common

evaluation metrics include BLEU (Papineni et al., 2002), ROUGE-1 (Lin & Hovy, 2003), ROUGE-L (Lin & Och, 2004), METEOR (Banerjee & Lavie, 2005), BERTScore (Zhang et al., 2019), and Hits@K. BLEU is primarily used for text generation tasks, such as machine translation. ROUGE-1 and ROUGE-L belong to the ROUGE family (Lin, 2004), originally designed for summarization evaluation. ROUGE-1 measures unigram recall between the predicted and reference summaries, while ROUGE-L considers the longest common subsequence between them. METEOR, originally developed for machine translation evaluation, focuses on string alignment. BERTScore measures the similarity between contextual embeddings generated by the BERT model (Devlin et al., 2019). Hits@k measures the percentage of test cases where the correct answer appears in the top $k$ predictions. For example, with the persona "I love hiking" and the correct response "I hike every weekend" ranked first, it counts as a hit for Hits@1, Hits@3, and Hits@5. Higher scores in these metrics indicate better model performance. However, there are scenarios where ground-truth text is not available. In such cases, model alignment with human pairwise preferences or reward models that capture user intent can serve as alternatives. However, this aspect has not been systematically explored yet.

- **Extrinsic Evaluation (Sec. 6.2)**: Extrinsic evaluation metrics can be expressed as $\psi(\mathcal{D}) = \{\mathbb{E}_e \mid \mathbb{E}_e(\hat{r}, r)\}$. These metrics are used to evaluate the generated content based on its effectiveness in downstream tasks, such as recommendation or classification. For recommendation tasks, common metrics include Recall, Precision, and Normalized Discounted Cumulative Gain (NDCG). In typical recommendation systems, the top-$k$ items are returned by personalized LLMs. Recall and Precision evaluate whether the predicted top-$k$ items match the expected top-$k$ items, while NDCG takes into account the ranking of the recommendations. For classification tasks, metrics such as Recall, Precision, Accuracy, and F1 Score are frequently used to measure performance.

Table 4 provides a taxonomy of evaluation metrics for personalized LLMs.

## 6.1 Intrinsic Evaluation

Intrinsic evaluation metrics are primarily used when ground truth textual data is available to assess the quality of generated content. In LaMP (Salemi et al., 2023), BLEU, ROUGE-1, and ROUGE-L are employed to evaluate models on tasks such as personalized news headline generation, personalized scholarly title generation, personalized email subject generation, and personalized tweet paraphrasing. More recently, the LongLaMP (Kumar et al., 2024) benchmark was proposed to evaluate personalized LLM techniques for longer-form personalized text generation. Similarly, ROUGE-1, ROUGE-L, and METEOR metrics are utilized to assess personalized LLMs on tasks like (1) personalized abstract generation, (2) personalized topic writing, (3) personalized review writing, and (4) personalized email writing. In addition, the win rate metric (Hu et al., 2024) is used to evaluate personalized responses for medical assistance (Zhang et al., 2023b), and Hits@K is applied to assess personalized responses in dialogues (Mazaré et al., 2018). Word Mover's Distance is another metric used to evaluate personalized review generation (Li & Tuzhilin, 2019). EGISES (Vansh et al., 2023) is the first metric explicitly designed to assess a summarization model's responsiveness to user-specific preferences. It does so by measuring the degree of insensitivity of model-generated summaries to variations across reader profiles, using Jensen-Shannon divergence to quantify the deviation between expected user-specific summaries and generated summaries. This approach allows EGISES to capture personalization independently of accuracy, establishing a baseline for evaluating how well a summarization model can adapt to individualized preferences beyond mere accuracy. Building on EGISES, PerSEval (Dasgupta et al., 2024) introduces a refined metric that not only assesses personalization but also integrates accuracy considerations to better reflect real-world performance. PerSEval differentiates itself by introducing the Effective DEGRESS Penalty (EDP), which imposes penalties for drops in summary accuracy and inconsistency across summaries. This design balances alignment with user preferences and accuracy, ensuring that high responsiveness does not mask poor accuracy, a limitation in EGISES.

LLMs are increasingly being used as evaluators to reduce the reliance on human labor. For example, MT-bench and Chatbot Arena (Zheng et al., 2023a) use strong LLMs as judges to evaluate these models on open-ended questions. However, questions remain about how reliably LLMs can serve as evaluators. Chiang & Lee (2023) first defined *LLM evaluation* as the process of feeding LLMs the same instructions and questions given to human evaluators and obtaining answers directly from LLMs. Judge-Bench (Bavaresco et al., 2024) provides

an empirical study comparing LLM evaluation scores with human judgments, concluding that LLMs are not yet ready to fully replace human judges in NLP tasks due to high variance in their evaluation performance. EvalGen (Shankar et al., 2024) incorporates human preferences by allowing users to grade LLM-generated prompts and code. This approach iteratively refines evaluation criteria based on user feedback, helping to validate the quality of generated content. In the context of personalization, to the best of our knowledge, there is still a lack of an LLM-as-a-judge framework specifically designed to systematically evaluate personalization applications. Developing such a framework or constructing robust reward models that accurately capture diverse user preferences could be a promising direction for future research. Additionally, human evaluation remains indispensable in this area. Methods like human preference judgments and pairwise comparisons are commonly used to assess the alignment between generated content and user-specific requirements. While LLMs can assist in the evaluation or annotation process (Li et al., 2023d), human evaluation remains essential for ensuring that personalized outputs truly meet user expectations, despite its practical cost.

## 6.2 Extrinsic Evaluation

Extrinsic evaluation metrics assess the quality of personalized LLMs in downstream tasks, such as recommendation and classification. For recommendation tasks, a common example is top-$k$ recommendation, where personalized LLMs predict which top-$k$ items to recommend. The predicted recommendations are then compared to the reference items (ground truth). Commonly used metrics include recall, precision, and NDCG. Recall measures the percentage of relevant items retrieved from the reference set, while Precision indicates the percentage of correctly retrieved items within the recommendations. NDCG evaluates how closely the ranking of recommended items matches the ground truth rankings. For other recommendation tasks, such as rating prediction, metrics like mean squared error (MSE), root mean squared error (RMSE), and mean absolute error (MAE) are frequently used.

In classification tasks, personalized LLMs classify input text (such as profiles or descriptions) into one of several candidate classes, which may involve binary or multiclass classification. For instance, personalized LLMs may match patients with suitable clinical trials, where the input includes patient health records and trial descriptions, and the output indicates whether there is a match (Yuan et al., 2023). In this context, metrics such as Recall, Precision, and F1 Score are used to evaluate the quality of the matches. These metrics are similarly applied in other classification tasks, including entity recognition and relation extraction (Tang et al., 2023). Additionally, metrics like accuracy and micro-F1 Score are often employed to assess the quality of classification outputs. For example, personalized LLMs may classify movies into specific categories based on their profiles (Salemi et al., 2023), with accuracy and micro-F1 used to measure classification performance. Beyond the tasks mentioned, a wide range of personalization tasks may require other task-specific metrics for evaluation. While most of these metrics often do not directly assess personalization, improvements often implicitly reflect enhanced personalization capabilities through better utilization of user information.

## 7 Taxonomy of Datasets for Personalized LLMs

We also categorize the datasets based on whether they are used for direct evaluation of personalized text generation through reward signals that reflect user preferences or for indirect evaluation in a downstream application. In the former case, as most existing datasets for personalized text generation rely on ground-truth text for evaluation, our discussion primarily focuses on these scenarios. In the latter case, the goal is to demonstrate that incorporating the generated text improves the performance of a downstream task, such as a recommendation model, classifier, or other functional systems.

- **Personalized Datasets *with* Ground-Truth Text (Sec. 7.1)**: Datasets containing actual ground-truth text written by users are relatively rare but essential, as they allow for direct evaluation of personalized text generation methods, rather than relying on performance in downstream tasks. Categories of datasets useful for direct evaluation of personalized text generation include **short-text generation** and **long-text generation** (Figure 5).

- **Personalized Datasets *without* Ground-Truth Text (Sec. 7.2)**: Datasets suited for indirect evaluation via downstream applications are far more common, as they do not require ground-truth text

Table 4: Taxonomy of Evaluation Metrics for Personalized LLMs

| Metric | Input format | Output format | Key Equation | Datasets or Domain |
|---|---|---|---|---|
| **TEXT-GENERATION** | | | | |
| **BLEU** | N-grams | Scalar | $BP \times exp\left(\sum w_n \log p_n\right)$ | Email, Amazon (Li et al., 2024b) |
| **ROUGE-1** | Unigrams | Scalar | $\frac{\|Y_i \cap \hat{Y}_i\|}{\|Y_i\|}$ | LaMP (Salemi et al., 2023) |
| **ROUGE-L** | Sequences | F-measure | $\frac{(1+\beta^2)Re \times Pr}{Re+\beta Pr}$ | LaMP (Salemi et al., 2023) |
| **METEOR** | Alignments | F-measure | $\alpha \times \frac{10 Re \times Pr}{Re+9Pr}$ | LongLaMP (Kumar et al., 2024) |
| **Bertscore** | Tokens | Scalar | (Zhang* et al., 2020) | Writing (Mysore et al., 2023a) |
| **Hits@K** | Answers | Scalar | $\frac{t \in \mathcal{K}_{test} \wedge t \leq k}{\|\mathcal{K}_{test}\|}$ | Dialogue (Mazaré et al., 2018) |
| **DOWNSTREAM TASKS** | | | | |
| **RECOMMENDATION** | | | | |
| **Recall** | Articles | Recommendations | $\frac{TP}{TP+FP}$@top-$k$ | Product, News (Ni et al., 2019; Wu et al., 2020) |
| **Precision** | Comments | Recommendations | $\frac{TP}{TP+FN}$@top-$k$ | Trip, Yelp (Li et al., 2020; Yelp., 2014) |
| **NDCG** | Descriptions | Rankings | $\frac{DCG}{IDCG}$@top-$k$ | Movie, Recipe (Lyu et al., 2023) |
| **CLASSIFICATION** | | | | |
| **Recall** | Profiles | Classes | $\frac{TP}{TP+FP}$ | Healthcare (Yuan et al., 2023) |
| **Precision** | Profiles | Classes | $\frac{TP}{TP+FN}$ | Citation (Salemi et al., 2023) |
| **Accuracy** | Profiles | Classes | $\frac{TP+TN}{TP+TN+FP+FN}$ | Category (Salemi et al., 2023) |
| **Micro-F1 Score** | Descriptions | Multi-Classes | $\frac{TP}{TP+\frac{1}{2}(FP+FN)}$ | Category (Salemi et al., 2023) |
| **LLMs AS EVALUATORS** | | | | |
| **IAA** | Prompts | Likert Scores | $\alpha = \frac{p_a - p_e}{1 - p_e}$ | Questions (Chiang & Lee, 2023) |
| **Correlations** | Texts | Ratings | $\frac{\sum(x_i-\bar{x})(y_i-\bar{y})}{\sqrt{\sum(x_i-\bar{x})^2 \sum(y_i-\bar{y})^2}}$ | Translation (Bavaresco et al., 2024) |
| **Pass Rate** | Prompts | Code | $\sum_{f \in F} PR(f) \times f(e)$ | Medical (Shankar et al., 2024) |

authored by users. These datasets are typically employed to evaluate personalized LLM techniques through tasks such as **recommendation**, **classification**, **dialogue**, and **question answering**.

Table 5 provides a comprehensive summary of various personalization tasks along with their key attributes. For each benchmark dataset, we indicate whether the data has been filtered to include only users with substantial prior activity (e.g., users who have reviewed at least 100 products, sent at least 100 emails, or rated a minimum of $k$ movies), which is relevant for addressing the cold-start problem. We also summarize whether the dataset contains user-written text, numerical attributes (such as ratings), and other categorical attributes, such as the genre of a movie watched. Additionally, we note if the dataset includes text descriptions (e.g., a movie's description) that are not written by the user but may still contribute to personalized LLM techniques, even though they are not user-specific.

## 7.1 Personalized Datasets *with* Ground-Truth Text

In terms of personalization datasets that actually contain the text written by users which can then be used for direct evaluation of the generated text fall under two main categories in our taxonomy shown in Table 5, notably, **short-text generation** and **long-text generation**. For instance, the review generation benchmark under long-text consists of all the reviews written by a user, the review title for each, and the ratings for every review as well, whereas most of the short-text generation datasets in Table 5 mostly consist of only the title of a news article or the title of an email. Notably, the output length column highlights the difference between **short-text generation** and **long-text generation** tasks. In particular, **personalized short-text generation** data seeks to generate very short text with a few words (*e.g.*, 9-10 words), and is somewhat similar to paraphrasing and summarization, as most of these datasets seek to generate a title of a paper, news article, email, among others. In contrast, data for benchmarking **personalized long-text generation** techniques is significantly much longer and thus more challenging as the goal is to generate a longer piece of

Table 5: **Taxonomy of Datasets for Personalized LLMs.** The datasets are categorized by the downstream task they have been applied to, including: short-text generation, long-text generation, recommendation, classification, dialogue, and question answering.. Notably, benchmark datasets for short-text and long-text generation contain user-specific ground-truth text, while others primarily rely on task-specific labels in different formats. For each dataset, we also indicate whether data has been filtered to include only users with a substantial amount of prior interactions (e.g., users who have reviewed at least 100 products, sent at least 100 emails, or rated at least $k$ movies). In addition, we identify whether the dataset contains user-generated text, numerical attributes (such as ratings), or other categorical attributes (e.g., the genre of a movie a user watched). While attributes like movie descriptions are text-based, they are not considered user-generated content for the purpose of personalized LLM techniques. Evaluation metrics are abbreviated as follows: ROUGE-1 (R1), ROUGE-L (RL), METEOR (MET), Precision (P), Recall (R), Normalized Discounted Cumulative Gain (NDCG), Hit Ratio (HR), Accuracy (ACC), Mean Absolute Error (MAE), Root Mean Square Error (RMSE), Perplexity (PPL), and additional metrics for opinion distribution analysis such as Representativeness, Steerability, Consistency, and Wasserstein Distance (Rep/St/Con/Was). "N/A" in the Output column indicates that the target is non-textual, while in the Evaluation Metrics column, it signifies the absence of established metrics for that dataset in personalization tasks, although the dataset may have potential for personalized applications.

| | Data | Size (users) | Output | Eval. Metrics | User Text | Numer. Attrs. | Cat. Attrs. | Timestamps | Filtered | |
|---|---|---|---|---|---|---|---|---|---|---|
| **SHORT-TEXT GENERATION** | News Headline | 13K/2K/2K | 9.7 | R1/RL | ✓ | | | ✓ | ✓ | Salemi et al. (2023) |
| | Scholarly Title | 10K/3K/3K | 9.2 | R1/RL | ✓ | | | ✓ | ✓ | Salemi et al. (2023) |
| | Email Title | 5K/1K/1K | 7.3 | R1/RL | ✓ | | | ✓ | ✓ | Salemi et al. (2023) |
| | Tweet Paraphrase | 10K/2K/1K | 16.9 | R1/RL | ✓ | | | ✓ | ✓ | Salemi et al. (2023) |
| **LONG-TEXT GENERATION** | Email Generation | 3K/1K/1K | 92 ± 60 | R1/RL, MET. | ✓ | | | ✓ | ✓ | Kumar et al. (2024) |
| | Abstract Generation | 23K/5K/5K | 160 ± 70 | R1/RL, MET. | ✓ | ✓ | ✓ | ✓ | ✓ | Kumar et al. (2024) |
| | Review Generation | 16K/2K/2K | 296 ± 229 | R1/RL, MET. | ✓ | ✓ | ✓ | ✓ | ✓ | Kumar et al. (2024) |
| | Topic Writing | 16K/2K/2K | 262 ± 241 | R1/RL, MET. | ✓ | ✓ | ✓ | ✓ | ✓ | Kumar et al. (2024) |
| **REC.** | MovieLens-1M | 6K (3.7K items) | N/A | P/R/NDCG | | ✓ | ✓ | ✓ | ✓ | Harper & Konstan (2015) |
| | Recipe | 2.5K (4.1K items) | N/A | P/R/NDCG | | ✓ | ✓ | ✓ | ✓ | Majumder et al. (2019) |
| | Amazon | 233M (15.2M items) | N/A | P/R/NDCG/HR | ✓ | ✓ | ✓ | ✓ | ✓ | Ni et al. (2019) |
| | Microsoft News | 1M (161K items) | N/A | P/R/NDCG/HR | | | ✓ | ✓ | | Wu et al. (2020) |
| | BookCrossing | 279K (271K items) | N/A | P/R/NDCG/HR | | ✓ | ✓ | | ✓ | Ziegler et al. (2005) |
| | TripAdvisor | 10K (6K items) | N/A | P/R/NDCG/HR | ✓ | ✓ | ✓ | ✓ | ✓ | Li et al. (2020) |
| | Yelp | 27K (20K items) | N/A | P/R/NDCG/HR | ✓ | ✓ | ✓ | ✓ | ✓ | Yelp. (2014) |
| **CLASSIF.** | MovieLens-1M | 6K (3.7K items) | N/A | P/R/NDCG | | ✓ | ✓ | ✓ | | Salemi et al. (2023) |
| | Movie Tag. | 4K/0.7K/0.9K | N/A | ACC/F1 | | ✓ | ✓ | ✓ | | Salemi et al. (2023) |
| | Product Rat. | 20K/2.5K/2.5K | N/A | MAE/RMSE | | ✓ | ✓ | ✓ | | Salemi et al. (2023) |
| **DIALOGUE** | ConvAI2 | 1.2K | 13.7 | PPL/F1/HR | ✓ | | | | | Dinan et al. (2020) |
| | Empathetic Conv. | 1.9K (79 personas) | N/A | N/A | ✓ | ✓ | ✓ | | ✓ | Omitaomu et al. (2022) |
| | PRISM | 8K | N/A | N/A | ✓ | ✓ | ✓ | ✓ | ✓ | Kirk et al. (2024) |
| **QUEST. ANS.** | OpinionQA | 1.5K | N/A | Rep/St/Con/Was | | | | | | Santurkar et al. (2023) |
| | GlobalOpinionQA | 2.6K | N/A | Similarity | | | ✓ | | | Durmus et al. (2023) |

text that is often 100s or 1000s of words. Nevertheless, all datasets under either of the proposed categories in our taxonomy, namely, short-text generation and long-text generation, can be used for directly evaluating the generated text as well as using the text provided for developing personalized LLM techniques via training or fine-tuning LLMs using the user-specific text provided for training.

## 7.2   Personalized Datasets *without* Ground-Truth Text

Personalized datasets that are useful for indirect evaluation of the generated text via a downstream application are by far the most common, as they do not require an actual set of ground-truth text written by individual users, and instead can leverage user attributes and other interaction data, to generate personalized text, and then this text is used to enhance another arbitrary model such as a recommendation approach. Such datasets that can be used in this fashion are shown in Table 5 in the following categories, including, **recommendation**, **classification**, **dialogue**, and **question answering**. Leveraging such category of evaluation strategy enables one to leverage any commonly used dataset that has been used for a variety of different personalization tasks such as recommendation, classification, and so on. However, a criticism of this approach is that we can only demonstrate that using the text is useful for the downstream application, but not that the text was actually meaningful or relevant to the user. On the other hand, for instance, generated intermediate tokens or embeddings can be used to augment user embeddings. The augmented embeddings are then shown to improve the performance of downstream tasks, highlighting the potential value of this approach while acknowledging its inability to validate user-specific relevance directly.

Additionally, we highlight datasets that, while not originally designed for personalization tasks, contain rich user-specific information and user-generated text, making them suitable for downstream personalization tasks. Notable examples include the PRISM Alignment Dataset (Kirk et al., 2024), which links survey responses and demographic information from diverse participants to their interactions with various LLMs, and the Empathic Conversations dataset (Omitaomu et al., 2022), which features multi-turn conversations enriched with detailed demographic and personality profiles, pre- and post-conversation data, and turn-level annotations. These datasets are valuable for tasks such as personalized QA, dialogue generation, empathy-aware responses, and user-specific emotion modeling in language models. We believe that although these datasets do not primarily use user-generated text as ground truth for evaluation, their rich user-specific information has the potential to capture diverse preferences. Developing effective reward models based on this information could be valuable for evaluating personalized text generation in cases where ground-truth user-generated text is unavailable, an area that remains largely unexplored.

# 8   Applications of Personalized LLMs

In this section, we explore various use cases where personalized LLMs have shown significant potential for enhancing user experiences and improving outcomes.

## 8.1   Personalized AI Assistant

### 8.1.1   Education

Personalized LLMs show promising potential to facilitate personalized education experiences for both students and teachers (Gonzalez et al., 2023; Wang et al., 2024b; Jeon & Lee, 2023; Huber et al., 2024; Wang & Demszky, 2023; Joshi et al., 2023; Yan et al., 2024b; Leong et al., 2024) and there have been increasing number of works (Sharma et al., 2023; Elkins et al., 2023; Ochieng, 2023; Olga et al., 2023; Phung et al., 2023) with such ideas. For example, they can analyze students' writing and responses, providing tailored feedback and suggesting materials that align with the students' specific learning needs (Kasneci et al., 2023). EduChat (Dan et al., 2023) tailors LLMs for educational applications by pre-training on educational corpora and stimulating various skills with tool use and retrieval modules through instruction tuning. It offers customized support for tasks such as Socratic teaching, emotional counseling, and essay assessment. Tu et al. (2023) use ChatGPT to create educational chatbots for teaching social media literacy and investigate ChatGPT's ability to pursue multiple interconnected learning objectives, adapt educational activities to users' characteristics (such as culture, age, and education level), and employ diverse educational strategies and conversational styles. While ChatGPT shows certain ability to adapt educational activities based on user characteristics with diverse educational strategies, the study identifies challenges such as the limited conversation history, highly structured responses, and variability in ChatGPT's output, which can sometimes lead to unexpected shifts in the chatbot's role from teacher to therapist. Park et al. (2024b) present a personalized tutoring system that leverages LLMs and cognitive diagnostic modeling to provide tailored instruction in English writing

concepts. The system incorporates student assessment across cognitive state, affective state, and learning style to inform adaptive exercise selection and personalized tutoring strategies implemented through prompt engineering. While their proposed system shows promise in adapting to individual students, the authors identified challenges in translating assessments into effective strategies and maintaining engagement, pointing to areas for further research in LLM-based personalized education. Overall, the challenges of personalized LLMs in education include copyright and plagiarism issues, biases in model outputs, over-reliance by students and teachers, data privacy and security concerns, developing appropriate user interfaces, and ensuring fair access across languages and socioeconomic backgrounds (Kasneci et al., 2023).

### 8.1.2 Healthcare

LLMs have demonstrated remarkable proficiency in various healthcare-related tasks (Liu et al., 2023e; Wang et al., 2023c; Liu et al., 2023h; Yang et al., 2024b), paving the way for their potential integration into personalized health assistance. Belyaeva et al. (2023) introduce HeLM, a framework that enables LLMs to leverage individual-specific multimodal health data for personalized disease risk prediction. HeLM employs separate encoders to map non-text data modalities (such as tabular clinical features and high-dimensional lung function measures) into the LLM's token embedding space, allowing the model to process multimodal inputs together. Abbasian et al. (2023) present openCHA, an open-source LLM-powered framework for conversational health agents that enables personalized healthcare responses by integrating external data sources, knowledge bases, and analytical tools. The framework features an orchestrator for planning and executing information-gathering actions and incorporates multimodal and multilingual capabilities. Building on openCHA, Abbasian et al. (2024) integrate specific diabetes-related knowledge to enhance performance in downstream tasks within the domain. Zhang et al. (2024a) introduce MaLP, a novel framework for personalizing LLMs as medical assistants. The approach combines a dual-process enhanced memory (DPeM) mechanism, inspired by human memory processes, with PEFT to improve LLMs' ability to provide personalized responses while maintaining low resource consumption. Jin et al. (2024b) propose a Health-LLM-based pipeline with RAG to provide personalized disease prediction and health recommendations. The system extracts features from patient health reports using in-context learning, assigns scores to these features using medical knowledge, and then employs XGBoost for final disease prediction.

### 8.1.3 Other Domains

Beyond the two domains where personalized LLMs have been widely employed, this section explores areas with less focus but significant potential for applying personalized LLMs. In these domains, specialized LLMs or language agent frameworks are emerging, but they often lack emphasis on personalization—an aspect that could greatly enhance user experiences.

**Finance**   Beyond the general advances of LLMs in finance (Araci, 2019; Wu et al., 2023b), personalized LLMs have shown significant potential in providing tailored financial advice, extending beyond general investment recommendations. For instance, Liu et al. (2023d) introduce FinGPT, a model that offers personalized financial advice by considering individual user preferences such as risk tolerance and financial goals. Additionally, Lakkaraju et al. (2023) evaluate the performance of LLMs as financial advisors by posing 13 questions related to personal finance decisions. The study highlights that while these LLM-based chatbots generate fluent and plausible responses, they still face critical challenges, including difficulties in performing numeric reasoning, a lack of visual aids, limited support for diverse languages, and the need for evaluation across a broader range of user backgrounds. Future applications of personalized LLMs in the financial domain could encompass a variety of specialized services. These may include personalized wealth management strategies, where LLMs offer dynamic advice on asset allocation and retirement planning, tailored risk assessment tools that provide custom risk profiles and real-time monitoring, and tax optimization strategies that help individuals and businesses minimize tax liabilities. Furthermore, LLMs could be deployed in personalized insurance solutions, credit management (including tailored loan recommendations and credit score optimization), and spending and budgeting tools that adapt to an individual's financial habits and goals. These applications could significantly enhance the relevance and utility of personalized LLMs in the financial sector.

**Legal**  A growing number of LLMs (Nguyen, 2023; Huang et al., 2023c; Cui et al., 2023) has been developed specifically for legal applications, where these models have proven useful in assisting judges with decision-making, simplifying judicial procedures, and improving overall judicial efficiency (Lai et al., 2023; Trautmann et al., 2022; Blair-Stanek et al., 2023; Yu et al., 2022; Nay, 2023; Fei et al., 2024). DISC-LawLLM (Yue et al., 2023) fine-tunes LLMs using legal syllogism prompting strategies and enhances them with a retrieval module to offer a broad range of legal services, potentially leveraging personal historical data. He et al. (2024b) introduce SimuCourt, a judicial benchmark comprising 420 real-world Chinese court cases, to evaluate AI agents' judicial analysis and decision-making capabilities. SimuCourt integrates AgentsCourt, a novel multi-agent framework that simulates court debates, retrieves legal information, and refines judgments using LLMs. This framework allows for the integration of different personas into various agents, enabling personalized interactions throughout the legal process. Looking ahead, we anticipate that personalized LLMs will significantly assist legal professionals by catering to their specific needs: For **lawyers**, personalized LLMs can be used for personalized case analysis, where they analyze legal cases in light of a lawyer's past cases, preferences, and typical strategies. This can lead to more effective argumentation and strategy formulation tailored to the lawyer's style. Moreover, personalized LLMs can enhance client interactions by adapting communication styles, content, and language to meet the unique needs of each client. This not only improves client satisfaction but also helps in maintaining long-term client relationships. Additionally, LLMs can assist in drafting legal documents, such as contracts and agreements, by incorporating specific clauses, language, and legal standards preferred by the lawyer or their firm. For **judges**, personalized LLMs can provide support in managing their caseload and ensuring consistent and fair rulings. Specifically, they can generate personalized case summaries that highlight the most relevant details based on a judge's past rulings and areas of focus, such as statutory interpretation or case law precedence. Furthermore, personalized LLMs can offer custom verdict recommendations that align with a judge's legal principles and prior decisions, promoting consistency in judicial outcomes. For **clients**, personalized LLMs can make legal services more accessible and tailored to individual needs. These models can offer personalized legal consultation by analyzing the client's specific circumstances, legal history, and goals, providing advice that is both relevant and easy to understand. Personalized LLMs can also provide clients with regular updates on their cases, offering clear explanations and progress reports that keep them informed and involved in the legal process. In summary, personalized LLMs have the potential to transform the legal domain by providing tailored support to different roles, enhancing efficiency, accuracy, and client satisfaction across the board.

**Coding**  As LLMs continue to advance, especially those fine-tuned on code-specific datasets (Roziere et al., 2023; Chen et al., 2021), their ability to generate high-quality code has seen significant improvement. This has spurred the development of an increasing number of AI-powered assistants aimed at enhancing the coding experience for programmers (Zhang et al., 2024b; Wang et al., 2024e;d; Xia et al., 2024). However, these applications often overlook the crucial aspect of personalization. For personalized code generation, Dai et al. (2024) propose MPCODER a novel approach for generating personalized code for multiple users that aligns with their individual coding styles. This method uses explicit coding style residual learning to capture syntax standards and implicit style learning to capture semantic conventions, along with a multi-user style adapter to differentiate between users through contrastive learning. The authors introduce a new evaluation metric called Coding Style Score to quantitatively assess coding style similarities. Personalization in coding assistance can be realized in several ways. Firstly, programmers and teams often have unique coding styles. For example, a middle-school student might prefer code that is easy to understand and well-commented, while a software engineer in a tech company may prioritize performance, scalability, and strict adherence to industry standards. A personalized LLM can learn from a user's behavior over time and adapt their suggestions to match the user's skill level, preferred frameworks, and commonly used coding patterns would significantly enhance their utility. Secondly, context-aware debugging is another area where personalization can make a substantial impact. Personalized LLMs could offer tailored debugging assistance based on a programmer's typical errors and preferred debugging strategies. Thirdly, enforcing code guidelines that align with a team's standards such as naming conventions, architectural patterns, and tool integrations is essential in collaborative environments. This ensures consistency and maintainability across the codebase, which is particularly critical in professional settings. Finally, personalized LLMs could greatly improve collaboration and code review processes by offering suggestions that consider both individual and team preferences. Achieving such levels of personalization requires advanced techniques like RAG or fine-tuning on user-specific data, enabling the model to adapt to

the distinct needs and preferences of different programmers. This represents a promising direction for future research and development in AI-powered coding assistants.

## 8.2 Recommendation

Personalized LLMs have been extensively applied across various recommendation tasks, including direct recommendations, sequential recommendations, conversational recommendations, explainable recommendations, rating predictions, and ranking predictions (Dai et al., 2023; Du et al., 2024; Hou et al., 2024; Liu et al., 2023a; 2024d; Ji et al., 2024b). These applications aim to enhance user experiences in recommendation domains like e-commerce (Tan & Jiang, 2023; Chen et al., 2024c). According to Wu et al. (2023a), the integration of LLMs into recommendation systems can be categorized into three main approaches: (1) augmenting traditional recommendation systems, such as collaborative filtering (Schafer et al., 2007; Resnick et al., 1994), with LLM embeddings; (2) enriching traditional recommendation systems by using LLM-generated outputs as features; and (3) employing LLMs directly as recommenders in downstream recommendation tasks. These three categories of approaches can all benefit from the personalization techniques discussed in Sec. 5. To enhance traditional recommendation systems with personalized LLMs, PALR (Yang et al., 2023) leverages LLMs to generate natural language user profiles, retrieve relevant item candidates, and rank and recommend items using a fine-tuned LLM. Chat-Rec improves traditional recommender systems by using LLMs to increase interactivity and explainability. It converts user profiles and historical interactions into prompts, allowing LLMs to learn user preferences through in-context learning and generate more personalized recommendations. For direct use of personalized LLMs as recommenders, InstructRec (Zhang et al., 2023a) frames recommendation tasks as instruction-following tasks for LLMs. It designs a flexible instruction format that incorporates user preferences, intentions, and task forms, and generates a large dataset of personalized instructions to fine-tune an LLM for recommendation tasks. Similarly, GeneRec (Wang et al., 2023b) employs LLMs for personalized content creation based on user instructions and feedback, aiming to complement traditional retrieval-based recommendation systems. While personalized LLMs have seen widespread application in recommendation systems, demonstrating superior performance in few-shot and zero-shot settings with enhanced explainability that addresses the cold-start problem (Liu et al., 2023a; Dai et al., 2023; Hou et al., 2024), significant challenges remain, including concerns over privacy, cost, and latency in large-scale deployments, highlighting the need for continued research and innovation

## 8.3 Search

Recently, with their growing capabilities in summarization and instruction following, LLMs have been incorporated into search engines (applications such as the new Bing (Microsoft, 2023) and SearchGPT (OpenAI, 2024)), which can provide an engaging conversational process that can help users find information more effectively (Spatharioti et al., 2023; Joko et al., 2024). Incorporating personalization can further tailor results to individual users' search histories, interests, and contexts, which can lead to more relevant and efficient search experiences (Bennett et al., 2012; Harvey et al., 2013; Cai et al., 2014; Song et al., 2014; Vu et al., 2014; 2015; Zhou et al., 2021). A large number of works (Dou et al., 2007; Sieg et al., 2007; Carman et al., 2010; Teevan et al., 2011; White et al., 2013) focused on how to better personalize search engines before the emergence of LLMs. In the era of LLMs, Zhou et al. (2024b) propose Cognitive Personalized Search (CoPS), a personalized search model that combines LLMs with a cognitive memory mechanism inspired by human cognition. CoPS utilizes sensory, working, and long-term memory components to efficiently process user interactions and improve search personalization without requiring training data. Besides, Jiang et al. (2024b) introduce Collaborative STORM, a system that personalizes search experiences by engaging users in multi-turn search sessions and incorporating their interaction history. However, despite the advancements in LLM-augmented search engines, there are still challenges to be addressed. Notably, Sharma et al. (2024) find that users of LLM-powered search exhibit more biased information querying and opinion polarization compared to conventional web search, especially when the LLM reinforces existing views. This phenomenon, known as the "echo chamber" effect (Pariser, 2011; Sharma et al., 2024; Lazovich, 2023; Garimella et al., 2018), emphasizes the challenges in balancing personalization with the need for diverse and objective information retrieval.

# 9 Open Problems & Challenges

Despite the significant progress made in the applications of personalized LLMs, there remain numerous unresolved challenges and open research questions. In this section, we explore key issues that require further investigation and innovation to advance the field. These challenges span various aspects of personalization, including the development of reliable benchmarks and evaluation metrics, tackling the persistent cold-start problem, addressing concerns about stereotypes and bias in personalized models, ensuring privacy in user-specific data handling, and expanding personalization to multi-modal systems. Each of these areas presents unique challenges that must be overcome to achieve more robust, fair, and effective personalized LLMs.

## 9.1 Comprehensive Evaluation: Benchmarks, Automatic Metrics, LLM-as-a-Judge, and Beyond

Effective benchmarks, combined with comprehensive metrics, are crucial for evaluating various aspects of LLMs, including their ability to personalize outputs. However, existing benchmarks for personalization are largely derived from recommendation systems, where the focus is predominantly on final predictions such as ratings, recommended items, or rankings. These benchmarks often overlook the intermediate processes in LLMs' output generation, which are critical for assessing whether the output is genuinely personalized. LaMP (Salemi et al., 2023) is one of the few benchmarks that specifically targets the evaluation of LLMs in generating personalized outputs. However, LaMP's scope is limited to text classification and short, single-turn text generation tasks. It lacks the complexity of real-world interactions, which are essential for applications like personalized AI assistants. This gap highlights the need for new benchmarks that can evaluate LLMs' personalized output generation in more realistic scenarios. Such benchmarks should also integrate personalization perspectives into other key LLM capabilities, including reasoning, planning, instruction following, and long-context understanding, thereby providing a more holistic evaluation. Overall, we envision comprehensive personalization benchmarks that effectively capture multi-turn interactions, evolving user preferences, and diverse contextual scenarios. For future directions in designing personalization benchmarks, one promising area involves language agents, which are increasingly applied to tasks such as web navigation (Deng et al., 2024), scientific discovery (Chen et al., 2024e; Si et al., 2024), research support (Asai et al., 2024), and coding assistance (Wang et al., 2024e). While their evaluation typically focuses on task success rates, the role of personalization in these contexts remains underexplored and calls for the development of dedicated evaluation criteria. Additionally, while current benchmarks predominantly emphasize English, there is an urgent need to expand their scope to include multilingual settings to ensure broader applicability and inclusivity. Given that LLMs are known to be sensitive to prompt variations (Zhuo et al., 2024), addressing distribution shifts and robustness in prompt design, as well as out-of-distribution scenarios (Wang et al., 2023a), becomes essential. Controlled experiments that systematically isolate and analyze dimensions such as style, topic, and user preferences can provide valuable insights. Furthermore, incorporating cultural and values-based adaptations (Shi et al., 2024b), alongside dialectal variations (Ziems et al., 2023), will enable these benchmarks to better reflect the real-world complexity of personalization and its diverse user base. Additionally, evaluating on static benchmarks often entails limitations, such as susceptibility to data contamination (Sainz et al., 2023; Balloccu et al., 2024) and a lack of adaptability to evolving LLM capabilities. To address these issues, designing dynamic evaluation frameworks for personalization tasks offers a promising alternative, enabling assessments with controllable complexity (Zhu et al., 2023b; Zhang et al., 2024c).

In addition, there is currently no comprehensive quantitative metric to assess the degree of personalization in LLM-generated outputs. Most existing metrics are task-specific and heavily dependent on the downstream task formulations and the quality of gold labels. As a result, they often fail to capture the diverse dimensions of personalization, such as those illustrated in Figure 4. The recent trend of using LLMs as judges in evaluating various aspects of LLM-generated content, due to their versatile nature, presents a promising approach for personalization assessment. Designing an LLM-as-a-Judge framework with personalized criteria rubrics could offer a more nuanced evaluation of the degree of personalization in LLM outputs. However, this approach remains underexplored, and challenges such as instability and potential biases need to be addressed to make it a reliable evaluation method. Specifically, challenges arise when LLM-based evaluators unintentionally inject their own internal persona or biases (Jiang et al., 2024a), and they can sometimes exhibit overconfidence by offering overly favorable assessments of their own outputs (Tian et al., 2023). Consistency is another issue (Li et al., 2024c), as LLM-as-a-judge evaluations often fluctuate depending on the context (Gu et al.,

2024) and exhibit phenomena such as positional biases (Li et al., 2024i), which poses significant problems in personalization tasks involving nuanced user attributes. There are also concerns about robustness, given that adversarial inputs can undermine the integrity of LLM-based evaluations (Doddapaneni et al., 2024; Raina et al., 2024; Shi et al., 2024a; Zheng et al., 2024). Furthermore, efficiency and flexibility can be limited when LLM-based evaluators must be manually prompted for each new personalization scenario. Despite these challenges, there are significant opportunities for improvement. Developing more refined prompting formats and strategies, such as scalar scoring, pairwise comparisons, and multiple-choice selections, can provide more nuanced and reliable evaluations. Additionally, addressing open problems like in-context exemplar selection could further enhance the evaluation process. Given the computational overhead of large-scale LLM-as-a-judge frameworks, which often rely on extensive API calls for model inference (Gu et al., 2024), training specialized smaller models tailored for personalization evaluation presents a promising alternative for reducing costs and improving efficiency (Kim et al., 2024; Huang et al., 2024a). Furthermore, exploring agent-as-a-judge paradigms (Zhuge et al., 2024) that incorporate multi-agent collaboration, tool integration, or human-in-the-loop approaches offers a path toward greater transparency, fairness, and robustness in assessing personalized outputs.

In summary, evaluating robust personalization in LLMs presents unique challenges compared to the evaluation of other capabilities. Personalization objectives in real-life applications are inherently diverse, leading to pluralistic alignment requirements (Sorensen et al., 2024). The varying levels of personalization defy reliance on any single evaluation criterion. Furthermore, the inherent ambiguity and subjectivity of user preferences add complexity, underscoring the urgent need for a finer-grained taxonomy to capture the full spectrum of personalization phenomena. Current datasets often suffer from imbalanced data, which can lead to incomplete or biased assessments of personalization performance. For instance, datasets may overrepresent users with frequent online activity, potentially skewing models to prioritize the preferences and behaviors of these users while underrepresenting less active user groups. Additionally, user preferences are dynamic and can shift over time, necessitating the development of multi-turn benchmarks that reflect evolving personalization goals and longitudinal variations. Moreover, due to inconsistencies in dataset structures, task formulations, and the diverse scenarios in real-life applications, direct performance comparisons can be challenging, making it difficult to achieve a unified and fair evaluation of different techniques. It is also critical to address privacy, bias, and fairness—factors that may conflict with certain personalization objectives—to ensure personalized systems remain safe, ethical, and equitable. Given these challenges, we advocate for collaborative efforts to create open-source platforms and shared tasks that allow personalized LLMs to be tested against standardized metrics aligned with core LLM capabilities. Finally, we emphasize the importance of cross-domain generalization tests beyond recommendation systems and text generation to evaluate how well personalization systems adapt across diverse domains and contexts.

## 9.2  Cold-start Problem

The cold-start issue is a prevalent and challenging problem in recommendation systems, where the system must generate recommendations for items that have not yet been rated by any users in the dataset, or when there is minimal information available about user preferences (Schein et al., 2002; Guo, 1997). Previously, a large number of methods (Lam et al., 2008; Li et al., 2019; Park & Chu, 2009; Lee et al., 2019; Wei et al., 2017) have been proposed to address such issues in traditional recommendation systems. Although LLMs demonstrate strong few-shot capabilities through in-context learning and role-playing via instructional prompting, significant challenges remain in effectively adapting personalized LLMs to sparse user-specific data via fine-tuning. This issue is further compounded by the fact that many downstream datasets are preprocessed to exclude instances with limited user interaction history—often filtering out data points where fewer than five interactions are recorded. As a result, the potential of personalized LLMs to handle low-resource scenarios remains relatively underexplored, and more advanced techniques are required to improve their adaptation to sparse data settings. Persona-DB (Sun et al., 2024) addresses cold-start problems more effectively through a hierarchical construction process that distills abstract, generalizable personas from limited user data. This is followed by a collaborative refinement stage that leverages similarities between users to fill knowledge gaps, allowing the system to draw relevant insights from users with richer interaction histories when personalizing for new or infrequent users. Given the limited number of works on personalized LLMs, we propose two potential research directions: (1) Building on the Persona-DB paradigm, through stages of prompting and

abstraction that progressively refine and generalize user personas, potentially improving personalization across diverse applications. (2) Leveraging synthetic data generation, which has shown promise in enhancing various LLM capabilities (Chan et al., 2024; Zhang et al., 2024c; Tong et al., 2024). LLMs could be employed to generate large-scale user-specific data from sparse seed data. However, challenges such as ensuring diversity and maintaining high-quality data at scale remain key obstacles to this approach.

### 9.3 Stereotype & Bias Issues of Personalization

The personalization of LLMs introduces significant concerns regarding the amplification and perpetuation of stereotypes and biases (Zhang et al., 2023d; Ziems et al., 2024; Gallegos et al., 2024; Li et al., 2023f). When LLMs generate personalized outputs, they rely on data that may inherently contain societal biases related to gender, race, ethnicity, culture, and other sensitive attributes. Personalization can unintentionally reinforce these biases by tailoring content that aligns with the biased data the models are trained on or the ones provided in the prompt, thus exacerbating the problem. For example, recent research (Gupta et al., 2023; Deshpande et al., 2023; Cheng et al., 2023a; Wan et al., 2023) indicates that assigning a specific persona to LLMs, like that of a "disabled person," can unexpectedly alter their performance across diverse tasks, including those seemingly unrelated to the assigned identity such as general knowledge or mathematical reasoning. Moreover, the feedback loop created by personalized systems can further entrench biases. Besides, Weissburg et al. (2024) examines bias in LLMs as personalized educators, showing disparities in content selection and difficulty based on demographics like race, gender, income, and disability. Using 17,000+ explanations and two bias metrics, it finds all tested LLMs exhibit bias, with income and disability status most affected. As LLMs continue to adapt based on user interactions, they might cater to pre-existing preferences and viewpoints, reducing the opportunity for exposure to diverse or corrective perspectives. This can lead to the deepening of echo chambers, where users are repeatedly exposed to biased or stereotypical information without opportunities for counterbalance. For example, Kantharuban et al. (2024) investigates how large language models generate recommendations that reflect both explicit and implicit cues of a user's identity, leading to biased outputs that align with racial stereotypes. The authors show that although personalization can tailor recommendations to the user's background, it often restricts the range of options for minority users and obscures the role of identity in the recommendation process. Despite growing efforts to mitigate biases in LLMs (Lu et al., 2020; Han et al., 2021; Ghanbarzadeh et al., 2023; Zhang et al., 2023d), there is a limited number of works on how personalization intersects with these biases. He et al. (2024a) introduce Context Steering (CoS), a training-free method for enhancing personalization and mitigating bias in LLMs at inference time by quantifying and modulating the influence of contextual information on model outputs. CoS works by computing the difference in token prediction likelihoods between models with and without context, then using this difference to adjust the influence of context during generation. However, CoS does not specifically aim to mitigate biases that arise within the personalization process. Vijjini et al. (2024) introduce the concept of personalization bias in LLMs, where LLM performance varies based on the user's demographic identity provided during the interaction. It proposes a framework to evaluate and quantify this bias by measuring safety-utility trade-offs across different user identities, using widely-used datasets. The study demonstrates the prevalence of personalization bias in both open-source and closed-source LLMs, and while it explores mitigation strategies like preference tuning and prompt-based defenses, it concludes that completely eliminating this bias remains an open challenge, highlighting the need for further research in this area. Overall, it is critical to design personalization systems that actively account for fairness and inclusivity, ensuring that personalized outputs do not reinforce harmful stereotypes or perpetuate biased perspectives. Future work should explore techniques such as bias detection during the personalization process, incorporating fairness constraints into the personalization pipeline, and ensuring that diverse perspectives are represented in user-specific outputs. For example, general LLM debiasing methods can serve as inspiration, including data-centric approaches such as data augmentation, filtering, reweighting training data, or modifying loss functions during fine-tuning to directly mitigate biases. Additionally, inference-time techniques—such as adjusting decoding strategies or rewriting model outputs—can help reduce bias in personalized responses while maintaining the model's tailored behavior (Gallegos et al., 2024). Addressing these challenges will require careful consideration of the trade-offs between personalization and fairness, as well as the development of robust evaluation frameworks that measure not only the degree of personalization but also the impact

on bias and stereotype propagation. Ultimately, ensuring that personalized LLMs promote inclusivity and fairness is essential to building systems that are socially responsible and beneficial to all users.

## 9.4 Privacy Issues

Privacy, particularly concerning Personally Identifiable Information (PII), is a critical concern in LLM personalization applications, where the objectives of personalization and privacy often conflict. Current LLMs are vulnerable to privacy breaches, as they can accurately infer personal attributes (e.g., location, income, gender) from unstructured text, even when common mitigations such as text anonymization and model alignment are employed (Staab et al., 2023). Additionally, adversarial attacks, such as prompt injections (Liu et al., 2023f; Zhan et al., 2024; Ning et al., 2024a) and jailbreaking (Xu et al., 2024b; Wei et al., 2024a), can cause LLMs to generate inappropriate content or reveal sensitive information from their training data. Although a growing body of research focuses on addressing privacy leakage in LLMs (Behnia et al., 2022; Lukas et al., 2023; Chen et al., 2023b; Yao et al., 2024; Yan et al., 2024a; Kuang et al., 2024; Feng et al., 2024), there is limited work specifically targeting the intersection of personalization and privacy (Zhao et al., 2024a). To address this gap, it is crucial to formally define the boundary between personalization and privacy, which may vary across tasks and can be subjective for different users. Furthermore, it is essential to design specialized modules that prevent both explicit and implicit privacy leaks throughout various stages of LLM personalization, such as data processing, model training, and retrieval processes. An ideal solution would allow for flexible adjustment, enabling a balanced trade-off between the degree of personalization and privacy protection, tailored to individual user preferences and specific application contexts.

## 9.5 Multimodality

Personalizing large multimodal models (LMMs) is particularly complex due to the diverse nature of the data they process, such as text, images, audio, and video. These models are designed to handle multiple input types and fuse them in meaningful ways to improve task performance across various domains. However, personalization in this context introduces unique challenges as it must account for individual user preferences or characteristics across multiple modalities simultaneously. In personalized multimodal models, the key challenge lies in effectively integrating user-specific data, such as preferences or interaction history, to modulate the model's responses in a contextually appropriate manner. These user-specific data points may come from different modalities as well. For example, in personalized image generation tasks, the model must be able to generate images that align with user-specific visual preferences while also understanding the associated textual or auditory cues. Recent work, such as Unified Multimodal Personalization (Wei et al., 2024b), demonstrates how LMMs can be adapted for personalization across multiple tasks and modalities. This framework unifies the personalization of tasks like preference prediction, explanation generation, and image generation under a common structure, leveraging the strengths of multimodal learning to predict user-specific outcomes based on a combination of text, images, and potentially other input types. Another notable example is the work on multi-subject personalization for text-to-image models (Jang et al., 2024), which focuses on personalizing generated images to represent distinct user preferences for multiple subjects within a single image. Similarly, techniques like personalized multimodal generative models (Shen et al., 2024) have been proposed to handle multiple modalities by transforming user behavior data into natural language for further personalization, extending the utility of LLMs in multimodal scenarios. Personalized multimodal models also pose significant computational challenges. The fusion of modalities requires more sophisticated architectures capable of jointly learning from multiple data streams without sacrificing personalization fidelity. For instance, plugging visual embeddings into traditional recommendation systems and personalized image generation models leverage LMMs to enhance the personalization process by embedding user preferences from both textual and visual input. To further push the boundaries of multimodal personalization, integrating modalities like video or audio into user-centric applications like recommendation systems and content generation presents another layer of complexity. Handling synchronization between modalities, ensuring cohesive user representations, and balancing personalization across dynamic content remain open challenges.

Specifically, according to Wu et al. (2024a), the unique high-level challenges in personalized multimodal LLM systems can be categorized as follows. (1) The integration of heterogeneous data. LMMs must process and align diverse data types such as text, images, audio, and structured user interaction history (Li et al.,

2024f). Encoding discrepancies arise as different modalities require distinct encoding techniques, making fusion difficult. Additionally, cross-modal alignment remains an issue since text and images may contain complementary but sometimes conflicting information. Another difficulty is data sparsity, where some users interact predominantly with one modality, leading to imbalanced personalization. Future opportunities include developing unified embedding spaces to bridge the gap between diverse modalities, leveraging self-supervised learning to enhance cross-modal understanding, and designing adaptive models that dynamically adjust weights for each modality based on user interaction patterns. (2) Data noise, redundancy, and quality control. Different modalities often include noisy, redundant, or irrelevant information (Liu et al., 2024e; Lyu & Luo, 2022). For example, images of the same object may vary in quality, while textual descriptions may be verbose or contain unnecessary details. Extracting meaningful insights while filtering out redundant or noisy data is essential for effective personalization. Future work should focus on implementing noise-aware training strategies to filter out irrelevant data during model training, using multimodal attention mechanisms to prioritize relevant user interactions and suppress redundant information, and employing contrastive learning to improve robustness against low-quality or inconsistent data. (3) Granular understanding of multimodal data. While text-based LLMs excel at linguistic processing, capturing subtle visual or auditory cues remains difficult (Shen et al., 2024). User preferences in areas such as fashion, art, or music often depend on fine-grained multimodal details, such as color, texture, or rhythm. Personalized multimodal models must enhance their ability to extract and relate these details meaningfully across different input types. Future advancements should develop hierarchical representations that preserve both fine-grained and high-level information, improve multimodal contrastive learning to align visual, auditory, and textual representations effectively, and explore fine-tuned retrieval-based techniques to improve nuanced personalization.

Beyond these overarching challenges, specific tasks introduce unique obstacles. In personalized image generation, achieving a balance between fidelity and diversity remains a key challenge. Hybrid diffusion models have shown promise in addressing this issue by refining subject representation and controlling generation constraints (Wang et al., 2024c; Ma et al., 2024; Song et al., 2025). Additionally, tokenization presents another challenge, as indexing multimodal inputs into lookup tables for efficient personalized generation requires specialized designs (Gal et al., 2022). For multi-modal personalized recommendation, multi-modal collaborative filtering introduces additional complexities. Recent approaches have attempted to unify multi-channel information, integrating generative recommendation mechanisms alongside dynamic item modifications (Yu et al., 2024). These methods enable tokenization of both items and user embeddings, facilitating multimodal feature incorporation into the model's latent space. Addressing these challenges will require continued research into improved multimodal alignment, data filtering, and scalable architectures that enhance personalization across diverse applications. Future advancements in integrating multimodal feedback loops and adaptive learning mechanisms will be essential for unlocking the full potential of personalized multimodal LLMs.

## 10  Conclusion

This survey provides a unified and comprehensive view of the burgeoning field of personalized LLMs. It bridges the gap between the two dominant lines of research — direct personalized text generation and leveraging personalized LLMs for downstream applications — by proposing a novel taxonomy for personalized LLM usage and formalizing their theoretical foundations. An in-depth analysis of the personalization granularity highlights trade-offs among user-level, persona-level, and global preference alignment approaches, laying the groundwork for future hybrid systems that can dynamically adapt to user needs and data availability. Additionally, this survey provides a detailed examination of techniques for personalizing LLMs, shedding light on the strengths and limitations of each approach. We explore the nuances of retrieval-augmented generation, various prompting strategies, representation learning approaches, and the evolving landscape of learning from personalized feedback through RLHF, underscoring the need for more robust and nuanced methods. Finally, our comprehensive survey of evaluation methodologies and datasets highlights the critical need for new benchmarks and metrics specifically designed for assessing personalized LLM outputs. Despite the significant progress made in personalized LLMs, numerous challenges and open problems remain. Key areas for future research include addressing the cold-start problem in low-resource scenarios, mitigating stereotypes and biases in personalized outputs, ensuring user privacy throughout the personalization pipeline,

and extending personalization to multi-modal systems. The field of personalized LLMs is rapidly evolving, with the potential to revolutionize human-AI interaction across diverse domains. By understanding the foundations, techniques, and challenges outlined in this survey, researchers and practitioners can contribute to the development of more effective, fair, and socially responsible personalized LLMs that cater to diverse user needs and preferences.

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
