# OpenReview forum: "Personalization of Large Language Models: A Survey"
_TMLR — Accepted by TMLR_

### Review · Reviewer_ctVC · 2025-01-03

**Summary Of Contributions:**

This survey provides a comprehensive overview of the key concepts, techniques related to personalizing LLMs. It examines two key areas of research: generating personalized text and using personalized information in downstream applications. The authors introduced taxonomy to unify, and a formalization of personalized LLMs. It surveys methods, metrics,  and outline critical challenges and open research questions in this area, such as benchmark development, addressing the cold-start problem, mitigating bias, ensuring privacy, and expanding to multimodal systems.

**Audience:**

Yes

**Claims And Evidence:**

Yes

**Requested Changes:**

1. **Critical**: Include practical examples and case studies to show how different personalization techniques are applied in real-world scenarios. This would improve the paper's relevance for practitioners seeking to implement personalized LLMs.
2. **Critical**: Provide deeper insights into evaluation challenges  for robust personalization evaluation.
3. **Strengthening**: Address computational and scalability concerns, with a specific section discussing trade-offs in implementing personalized LLMs, since it might limit what practionnerrs can use in real-world applications.
4. **Strengthening**: Expand the discussion on using LLMs as evaluators for personalized LLM outputs. Address potential challenges and opportunities in this area, providing insights for future research.
5. **Strengthening**: Provide a more in-depth analysis of the challenges and opportunities in developing personalized multimodal systems. This is a rapidly evolving area that deserves further attention.
6. **Critical**: Provide comparative analyses or benchmarks of existing personalization datasets and techniques to guide practitioners.

**Strengths And Weaknesses:**

**Strengths:**

1. Overall, this paper is a valuable contribution to the field of personalized LLMs.
2. The paper provides a thorough understanding of the current state of the field of personalized LLMs.
3. Proposes a unifying view, taxonomy and formal definitions for personalized LLMs which could provides guideline for future work to ground the terminology,  clarity and unification for future work in this field.

**Weaknesses:**

1. Limited Practical Examples: While the paper includes a theoretical overview, it would benefit from including more practical examples and case studies to illustrate the concepts and techniques discussed, such as computational overhead and scalability of the proposed techniques.
2. Depth of Discussion on Emerging Topics: Some emerging topics, such as the use of LLMs as evaluators and personalized multimodal systems, could benefit from a more in-depth discussion, considering their growing importance in the field.
3. Bias and Ethical Concerns: While briefly mentioned, the paper does not deeply analyze mitigation strategies for biases in personalized outputs.
4. Technical Depth: advanced techniques like RLHF, could benefit from more technical details.
5. Method Comparison: The work does not compare exiting methods on existing benchmark to contrast their performance.

---

> ### Author Response · Authors · 2025-03-10
> **Response to Reviewer ctVC Part 1**
>
> We sincerely appreciate your constructive feedback and your **recognition of our survey as a valuable contribution to the field of personalized LLMs**. We are grateful for your acknowledgment of our **comprehensive overview, the unifying taxonomy, and the formalization of personalized LLMs**, which help provide clarity and structure for future research. We address your concerns below and summarize corresponding improvements (highlighted in blue in the paper).
>
> - **W1 / Requested Change 1: Limited Practical Examples**
>   Thank you for this suggestion. In the original paper, we included Figure 5 to illustrate personalized LLM tasks, we acknowledge that it lacks concrete demonstrations of how different personalization techniques are applied. However, it lacked concrete examples of how different personalization techniques are applied. To address this, we have **added Figure 7**, which presents a **case study** on personalized movie review generation. This figure provides a detailed example of how various LLM personalization techniques (e.g., retrieval-augmented generation, prompt engineering, and fine-tuning) are applied in practice, offering better clarity on real-world applications.
>
> - **W2 / Requested Changes 2 & 4: Deeper Insights into Evaluation Challenges and LLMs as Evaluators**
>   Thank you for bringing up the need for a more thorough analysis of evaluation challenges in LLM-based personalization and addressing the role of LLMs as evaluators. To address this, we have updated **Section 9.1 (page 37)** as follows:
>
>   - We have **expanded the discussion on evaluation challenges**, emphasizing the importance of realistic benchmarks, comprehensive metrics, dynamic evaluation frameworks, and the need for a deeper taxonomy of personalization phenomena.
>   - We have **added a detailed analysis of the opportunities and limitations of using LLMs as evaluators** for personalized outputs, including concerns about biases, instability, and efficiency.
>   - We have integrated perspectives on **future directions**, including language agents in various application domains, multilingual evaluations, privacy and fairness considerations, and cross-domain generalization tests.
>
>   We believe this additional content directly addresses your concern and hope that these revisions provide a clearer understanding of the evaluation landscape for personalized LLMs and the critical factors that influence reliable and ethical assessment.
>
> - **W3: Lack of In-Depth Analysis of Bias Mitigation Strategies in Personalization**
>   We agree that bias mitigation in LLM personalization is an important topic. However, as far as we know, there are currently no existing works specifically addressing this issue. As a result, our survey, which focuses on general LLM personalization, highlights this gap and calls for future research in **Section 9.3**. To further address your concerns, we have incorporated recent studies analyzing biases in personalized LLMs and expanded our discussion on relevant debiasing techniques from general LLM applications (highlighted in blue in **Section 9.3 on Page 39**).
>
> - **W4: Lack of Technical Depth in Advanced Techniques like RLHF**
>   Thanks for this comment! We want to respectfully point out that our survey focuses on general LLM personalization and not an in-depth technical analysis of specific methods. For example, in **Section 5.4**, we have already provided a comprehensive discussion of RLHF in the context of personalization. While we acknowledge the importance of technical details, covering every advanced technique in depth is beyond the scope of this survey, given the broad range of methods involved in LLM personalization. However, we believe that by defining key foundational concepts, we have provided readers with a strong framework to understand these techniques.

---

> ### Author Response · Authors · 2025-03-10
> **Response to Reviewer ctVC Part 2**
>
> - **Requested Change 3: Computational and Scalability Concerns**
>   We agree that computational efficiency and scalability are critical considerations for practitioners. To address this, we have added a new **Section 5.5 on Page 27 (Discussion) (highlighted in blue)**, where we explicitly discuss the trade-offs in implementing personalized LLMs. This section highlights key factors such as:
>
>   - The resource overhead of retrieval-based methods (RAG, hybrid sparse-dense approaches).
>   - The latency implications of prompting-based solutions.
>   - The potential of distillation-based models, caching pipelines, or on-device personalization to reduce computational costs.
>   - The challenges of gathering sufficient feedback for RLHF and alternatives like RLAIF.
>
>   Ultimately, we emphasize that practitioners must balance personalization gains against infrastructure constraints and practical feasibility.
>
> - **Requested Change 5: Personalized Multimodal Systems**
>   We acknowledge the growing importance of personalized multimodal systems and have expanded our discussion in **Section 9.5 on Pages 40-41 (highlighted in blue)** to address complexities unique to personalizing large multimodal models. Newly added content covers key challenges such as:
>
>   - Integrating heterogeneous data (text, images, audio, video) while maintaining alignment across modalities.
>   - Filtering out noise and redundancy to ensure high-quality personalization.
>   - Capturing fine-grained user preferences.
>   - Task-specific hurdles in areas like personalized image generation, specialized tokenization for multimodal inputs, and multi-modal collaborative filtering.
>
>   These additions underscore the need for advanced alignment, adaptive learning mechanisms, and scalable architectures to fully realize the potential of personalized multimodal systems.
>
> - **W5/ Requested Change 6: Comparative Analyses or Benchmarks of Existing Personalization Datasets**
>   We agree that comparing different personalization datasets is important due to their varying focuses and attributes. In response, we respectfully highlight that our survey already includes a detailed comparison and taxonomy of personalization datasets in **Section 7**, with **Table 5** explicitly comparing datasets based on multiple aspects such as size, output format, evaluation metrics, and key personalization attributes. We believe this provides a structured reference for researchers and practitioners.
>
>   Regarding benchmarking personalization techniques across datasets, we acknowledge its value but also note that:
>
>   - Due to inconsistencies in dataset structures and task formulations, direct performance comparisons can be challenging.
>   - Personalization methods are subject to constraints such as efficiency, accessibility, and domain specificity, making universal benchmarking non-trivial.
>
>   Nevertheless, we have clarified these limitations in our discussion to ensure transparency. We have incorporated this into **Section 9.1 on Page 37** as part of the challenges in evaluating personalized LLMs.
>
> ---
>
> We appreciate your insightful feedback, which has helped us improve the clarity, depth, and practical relevance of our survey. Thank you again for your time and thoughtful review!

---

> > ### Comment · Reviewer_ctVC · 2025-03-14
> >
> > Thank you for your revisions and for carefully addressing all the critical comments from the previous round of review. The responses provided were comprehensive, and the additional details added have significantly enhanced the quality of the paper. I have no further concerns.

---

### Review · Reviewer_NNZE · 2025-02-06

**Summary Of Contributions:**

In this paper, the authors survey the topic of personalization of Large Language Models (LLMs), highlighting their growing importance and applications. ​The authors attempt to unify personalized text generation and downstream task personalization and introduce a taxonomy for personalized LLM usage. ​The survey formalizes foundational concepts, discusses personalization granularity, and categorizes techniques like retrieval-augmented generation, prompt engineering, and reinforcement learning from human feedback. ​ It also reviews evaluation metrics, datasets, and applications in education, healthcare, finance, legal, and coding. The document identifies challenges such as the cold-start problem, biases, privacy concerns, and the need for improved benchmarks and metrics, emphasizing the potential for future research and innovation. ​Overall it is a strong introduction to the topic.

**Audience:**

Yes

**Broader Impact Concerns:**

Not applicable.

**Claims And Evidence:**

Yes

**Requested Changes:**

Look at my weakness pointed out.

**Strengths And Weaknesses:**

Strengths:

The authors provide an extensive overview of the personalization of LLMs, covering various aspects such as techniques, evaluation metrics, datasets, and applications.
I appreciate the authors' attempt at keeping the paper easily accessible by carefully defining the terminologies used, and remaining consistent throughout.
The taxonomy is well-structured taxonomy for personalized LLM usage.
The challenges and open problems discussed are interesting covering a variety of topics from benchmarking, cold-start, privacy issues to multi-modality.

Weaknesses: ​

Although the  authors emphasize on bridging personalized text generation and downstream task personalization as the motivation for the survey paper, I find this emphasis a bit heavy for a reader looking for an introduction to the topic.
I suggest that this emphasis can be toned down or rather reworded to point at the reader to the larger picture of introducing the LLM personalization.

---

> ### Author Response · Authors · 2025-03-10
> **Response to Reviewer NNZE**
>
> We sincerely appreciate your valuable feedback and the acknowledgment of our **extensive overview of LLM personalization, well-structured taxonomy, and the careful definition and consistency of terminology** throughout the paper. We are also grateful for your recognition of our discussion on key challenges, including benchmarking, cold-start issues, privacy concerns, and multimodality, as well as your appreciation of our survey as a **strong introduction** to the topic. Below, we provide clarifications to address your concerns:
>
> ### **W1: Too Strong Emphasis on Bridging Personalized Text Generation and Downstream Task Personalization for Some Readers**
> - Thank you for pointing this out! We acknowledge that our introduction initially placed a strong emphasis on bridging these two directions to highlight gaps in prior work and limitations in existing surveys. To better address your concern, we have **refined our paper** by compressing this part and repositioning it as an introduction to the broader topic of LLM personalization. Specifically, inspired by Reviewer K7BN's suggestion, **we have reframed it as a starting point** and emphasized that as LLMs continue to evolve, these two directions will increasingly converge, motivating a more universal perspective on personalization.
> - The revised section (highlighted with blue on **page 2**) explicitly emphasizes the need for a unified framework while ensuring accessibility by expanding the discussion to foundational concepts, techniques, datasets, evaluation methods, and real-world challenges. This restructuring provides a more balanced introduction without overemphasizing any single aspect. We appreciate your insightful feedback, which helped us improve the clarity and scope of our survey.

---

> > ### Comment · Reviewer_NNZE · 2025-04-03
> > **Official comment by Reviewer NNZE**
> >
> > I appreciate the authors' efforts to addressing my comments from my review. The reworded introduction coherently bridges the topic. I have no further concerns.

---

### Review · Reviewer_K7BN · 2025-02-26

**Summary Of Contributions:**

This paper offers a comprehensive survey on the state of the art in LLM personalization.  In particular, their approach is inspired by the dual-track nature of LLM personalization research: works are focused on either (1) personalized text generation, or (2) using LLMs in the service of some other downstream task, and the relationship between these directions has not been well characterized.  First, the authors describe these settings and offer the formalizations necessary to describe these settings mathematically.  Then, taxonomies are offered for: (a) levels of granularity of personalization (i.e., individual vs. demographic group vs. population) (b) methods for personalization (c) metrics and evaluation (d) datasets. Finally, applications and open challenges are discussed.  Many relevant and recent references are cited throughout the paper.

**Audience:**

Yes

**Claims And Evidence:**

Yes

**Requested Changes:**

- Notation: Start with most general cases, introduce special cases like RAG or particular evaluation paradigms later as special cases building on original general notation.  Streamline where possible, dropping redundant notation.  Make sure reward models, preference data, pairwise feedback, etc. are incorporated seamlessly.
- Improve Section 7 on dataset taxonomy to incorporate the cases where there is no ground truth user-written data, but the goal is not a downstream separate task like recommendation.
- Make all changes everywhere else needed to clarify the role of preference data/reward models.
- Make Figure 1 more general, remove special cases like RAG and user ground truth written text.
- Section 3.5 asserts that the taxonomy given is comprehensive.  This needs to be better justified, through argument and/or citation.

**Strengths And Weaknesses:**

## Strengths

This work offers a useful high-level characterization of an important field of research.  I think that we’ve only seen the tip of the iceberg in research in LLM personalization, and this study comes at an exciting time for the field.

I think noting the dual-track nature of LLM personalization is very useful.  I have tried to research LLM personalization before, and had to put considerable effort wading through papers of one type (in my case, recommendations) when I was actually looking for the other (text generation).  This survey can stand as a valuable resource for researchers aiming to understand what research might fall under the heading of LLM personalization, and home in on the particular research they are looking for.

## Weaknesses

As I mentioned previously, I find the dichotomy noted between text generations and recommendation systems to be very useful.  However, I think the authors could take their motivation for this further.  Instead of noting that these are somewhat separate topics and should be treated as such, I think the logical end here is that while these have been treated as separate topics because of technical limitations of our current models, ultimately they will merge.  For example, a single language model can be responsible for holding free form text conversations with a customer, and then reasoning over text entries in a company catalog to recommend items.  It is worth accurately characterizing and unifying these two separate research tracks, so that we may realize a future where a single intelligent agent orchestrates an end-to-end user experience.

While the previous point is more opinion-based, I think the paper has some technical weaknesses that need to be addressed in order to advocate for acceptance. First, I think the structure(s) created by the authors in the paper do not neatly handle the case of RLHF and preference data, which I think covers many or most of the interesting use cases going forward.  For example, Figure 1, and in particular page 5, describe evaluation of personalized text generation as involving actual text written by that user (or user-written ground truth, etc.).  However, for many personalized text generation tasks, this does not make sense.  Consider the example of the personalized mental health chatbot given on the bottom of page 3.  That chatbot would clearly not be scored for matching a text written by the user, if for no other reason than because the user is not a qualified professional (which also maps onto domains like medicine and education).  Instead, it would be scored by some reward model representing the user’s preferences over expert responses, which may be derived from existing pairwise feedback from the user (or other similar users).  I think this paradigm will describe much of the future research on LLM personalization, and failing to capture it cleanly weakens the paper.

I think this problem especially affects Section 7.  The development of many personalized LLM applications will not involve ground truth data written by the user.  However, this section then equates not having ground truth user-written text with evaluation via downstream applications.  This section, as it stands, seems incorrect/incomplete.

Next, I think the formalization needs significant work.  I think the formalization in Sections 2 and 3 should be more general, and the specific techniques and settings described in later sections should be special cases.  For example, I think it is wrong to focus on RAG in equation (1) and (3).  This section should be very general statements about personalization, and RAG and other approaches should be described as special cases later in Section 5.  Other notes with notation:
 - In section 3.1, isn’t the output \hat{Y} a text sequence as well?
 - Again in section 3.1, calling \mathcal{F} a task seems odd.  This is a model, correct?  A task does not typically produce a prediction.
 - How does Definition 3 relate to the notation previously introduced? How does \mathcal{H} relate to X?
 - None of the definitions in 3.2 capture pairwise feedback on text generations well
 - Definition 4 indexes users by i, and then Definitions 8-10 use u.
 - I found the notation in Section 3.4 to be cumbersome, and thus I struggled to understand what the takeaway from that section should be.
 - There is a lot of notation still being introduced in Sections 4 and 5, I think this should either be introduced up front, or much of it may be redundant and should be re-expressed according to variables introduced earlier.
- Section 6 on evaluation of text generations does not capture many relevant scenarios governed by reward models (see above)

Finally, Figure 1 should be more general (e.g., w.r.t. RAG, evaluation of ground truth user written text, only using data from the specific user).  Also, the blue box should probably only contain the elements that are specific to personalized text generation, not those shared by both paradigms.

As it stands, I think the paper’s weaknesses outweigh its strengths, and I will respond NO to claims/evidence.  However, I think the contribution could be valuable and the paper could be pushed over the acceptance threshold with the below changes.

---

> ### Author Response · Authors · 2025-03-10
> **Response to Reviewer K7BN Part 1**
>
> We sincerely appreciate your thoughtful review. We appreciate your recognition of our survey’s contributions, particularly in **characterizing the dual-track nature of LLM personalization** and providing a **structured taxonomy** to help researchers navigate this evolving field. We are also glad for your appreciation of our work **coming at a pivotal moment for the field**. We address your concerns below and summarize corresponding improvements (highlighted in blue in the paper).
>
> - **W1: Insufficient motivation for separating the two personalization tracks**
>   Thank you for highlighting this important point. We agree that explicitly clarifying the logic behind discussing the two personalization tracks separately in the current state is necessary. We welcome your insightful suggestion to better motivate the relationship between these tracks as currently separate but potentially converging research directions. Based on your feedback, we **revised the second-to-last paragraph preceding the contributions in Section 1 on Page 2**.
>
> - **W2: Inadequately addresses RLHF and preference-based personalization that does not rely on user-written ground truth texts**
>   Thanks for pointing this out. We acknowledge that many scenarios lack ground-truth user-written text, and in our original paper, we highlighted that personalized text generation remains particularly challenging due to the **scarcity of datasets with high-quality user-written labels in Section 2**. Additionally, **most existing research**
>
> >> LaMP: When Large Language Models Meet Personalization
>
> >> LongLaMP: A Benchmark for Personalized Long-form Text Generation
>
> on personalized text generation **relies heavily on ground-truth data for evaluation**, with more established evaluation methods in this setting. Therefore, we place greater emphasis on scenarios where ground-truth text is available.
>
>   To further address your concerns, we **have revised our wording** to consider the scenarios when ground-truth text is not available when evaluating personalized text generation. In particular, we have replaced **"ground-truth text"** with a more general term, clarifying that evaluation can be based on user-written text when available or, alternatively, on user preferences and separate reward models that reflect user judgments, as seen in the caption of **Figure 1**. Please refer to the detailed responses below to review our revisions in accordance with your mentioned points. (Changes are made in **Section 2 on Pages 4-5, Figure 1, Section 6 on Pages 28-29, and Section 7 on Pages 30-33**.)
>
> - **W3: Formalization needs significant work**
>   - **(a) More general notations in Sections 2, 3 and specific in Section 5**
>     Thank you for pointing this out. To address your concerns, we have updated the notations in **Sections 2 and 3**, replacing the RAG-related notations with a more general adaptation function in places such as **Equation 1 on Page 4** and **Figure 1** (highlighted in blue). Additionally, we still use task-specific notations in Section 5.
>
>   - **(b) Other specific notation questions:**
>     - **In Section 3.1, is the output \(\hat{Y}\) a text sequence as well?**
>       Yes, it represents personalized text. You can refer to **Table 2** for details. Additionally, we have added an explanation in **Section 3.1** (highlighted in blue).
>     - **In Section 3.1, calling \(\mathcal{F}\) a task seems odd. This is a model, correct? A task does not typically produce a prediction.**
>       We agree with the reviewer that a task itself does not directly produce predictions. To clarify, we have revised **Definition 3.1** so that a downstream task explicitly refers to a practical application or goal (e.g., classification, translation), whereas the notation \(\mathcal{F}\) specifically denotes a downstream model or a sequence of operations applied to the model-generated outputs to achieve the task.
>     - **How does Definition 3 relate to the notation previously introduced? How does \(\mathcal{H}\) relate to \(X\)?**
>       Definition 3 provides a general definition for an LLM input for a chat-style input for LLMs, which in most cases contains a system prompt and user prompts. As **clearly stated in the last sentence of Definition 3**, **user prompt** \(\mathcal{H}_{user}\) is equal to \(x\) in the following section.
>     - **None of the definitions in Section 3.2 capture pairwise feedback on text generations well.**
>       We have updated **Definition 5 on Page 7** to include pairwise feedback and are open to further revisions if you can specify which other definitions should explicitly incorporate it.

---

> ### Author Response · Authors · 2025-03-10
> **Response to Reviewer K7BN Part 2**
>
> - **Indexes users by \(i\), and then Definitions 8-10 use \(u\).**
>       Thanks for pointing this out! We have revised the user index in **Definition 4** using \(u\) instead of \(i\) for consistency (highlighted in blue).
>     - **I found the notation in Section 3.4 to be cumbersome, and thus I struggled to understand what the takeaway from that section should be.**
>       Thanks for the feedback. We have revised the **Figure 3 caption** to be consistent with the notation used in **Section 3.4 on Page 10**, and have better highlighted the key takeaways in the main text of **Section 3.4**.
>     - **There is a lot of notation still being introduced in Sections 4 and 5. I think this should either be introduced upfront, or much of it may be redundant and should be re-expressed according to variables introduced earlier.**
>       Thank you for pointing this out. We introduce the formal definitions of terms such as *Personalization Granularity*, *Retrieval Model*, *RAG*, and other task-specific concepts in Sections 4 and 5 rather than upfront for the following reasons:
>       1. Sections 2 and 3 focus on **foundational concepts and a general introduction** to Personalized LLMs, rather than diving into the technical details of specific techniques and models.
>       2. These sections already introduce many definitions, and adding additional technical notation here could **overwhelm the reader**. Instead, we introduce terms at the points where they become most relevant to the discussion.
>
>       We welcome further suggestions on the placement of these definitions and are open to revising the manuscript accordingly.
>     - **Section 6 on evaluation of text generations does not capture many relevant scenarios governed by reward models (see above).**
>      To address your concerns, we have added additional content on pairwise preferences and reward models for evaluation on **pages 28-29**, highlighted in blue, to further emphasize this point.
>
> - **Requested Change 1: Notation – Start with most general cases, introduce special cases like RAG**
>   Thanks! We have updated the notations in **Sections 2 and 3**, replacing the retrieval module with a general adaptation function (\(A\)) that captures user-specific information, for example in **Equation 1 on Page 4**. Additionally, we have ensured that technique-specific notations appear only in their corresponding sections.
>
> - **Requested Change 2: Improve Section 7 on dataset taxonomy to incorporate cases where there is no ground truth**
>   We have added additional discussion on scenarios where ground-truth text is unavailable, emphasizing this setting on **Pages 31-32**. Additionally, at the end of this section on **Page 33**, we highlight that existing datasets without ground-truth text have the potential to support the development of effective reward models, which could be valuable for evaluating personalized text generation (highlighted in blue). As far as we know, there are currently no datasets for personalized text generation that explicitly cover cases where ground-truth text is unavailable. We are open to further modifications if you have additional suggestions.
>
> - **Requested Change 3: Make all changes everywhere else needed to clarify the role of preference data/reward models.**
>   We have already made all the necessary changes to clarify this and are open to further modifications if you have additional suggestions.
>
> - **Requested Change 4: Make Figure 1 more general**
>   Thank you for your feedback. We have updated **Figure 1** by replacing the **Retrieval module** with a more general **Adaptation Function**, which accounts for various techniques used to incorporate user-specific information in personalized LLMs. We have introduced a separate **Figure 7**, which provides a detailed illustration of these adaptation techniques, including **RAG, fine-tuning, etc.**, with concrete examples. Furthermore, we have **revised the caption** to emphasize that the evaluation of personalized text generation can be based on preference-formatted data or reward models, rather than solely relying on user-written text.
>
> - **Requested Change 5: Section 3.5 asserts that the taxonomy given is comprehensive. This needs to be better justified, through argument and/or citation.**
>   Thanks, we have now revised this on **Page 10** to avoid mentioning that it is comprehensive.

---

> > ### Comment · Reviewer_K7BN · 2025-03-11
> > **reviewer reply**
> >
> > Thank you to the authors for considering my feedback and updating their manuscript.  My main concerns have largely been addressed, and I have revised my score on Claims and Evidence to "yes".  I think this is a solid survey that can serve as a resource for researchers aiming to get a picture of the LLM personalization landscape.

---

> > > ### Author Response · Authors · 2025-03-11
> > > **Response to Reviewer K7BN 's response to author's rebuttal**
> > >
> > > Thank you so much for taking the time to review our rebuttal and for recognizing the updates we made! We appreciate your positive feedback and are glad to hear that you find our survey a valuable resource for researchers in LLM personalization.

---

### Decision · Action_Editor_GPso · 2025-05-27

**Recommendation:** Accept as is

**Comment:**

This is a detailed survey, and all the reviewers are satisfied with the state of the survey after the rebuttal. They all agree to accept the paper. Given the detailed and timely nature of this survey, I recommend a survey certification for this paper.

**Audience:**

This paper will be of interest to anyone working in LLMs and their applications, which is the majority of the ML field right now. Hence, the survey is valuable for several members of the TMLR audience.

**Claims And Evidence:**

This paper provides a comprehensive survey on the personalization of LLMs. The authors focus on two major areas of personalization: personalized text generation and using personalized information in downstream applications. All the reviewers agree that the survey is very comprehensive, well written and useful to the community!